# Structured Energy Network as a Loss Function

**Jay-Yoon Lee**[¶]
lee.jayyoon@snu.ac.kr

**Dhruvesh Patel**[♣]
{dhruveshpate,

**Purujit Goyal**[♣]
purujitgoyal}@cs.umass.edu

**Wenlong Zhao**[♣]
wenlongzhao@cs.umass.edu

**Zhiyang Xu**[◇]
zhiyangx@vt.edu

**Andrew McCallum**[♣]
mccallum@cs.umass.edu

[¶]Graduate School of Data Science, Seoul National University
[♣]Manning College of Information & Computer Sciences, University of Massachusetts Amherst
[◇]Department of Computer Science, Virginia Tech

## Abstract

Belanger & McCallum (2016) and Gygli et al. (2017) have shown that energy networks can capture arbitrary dependencies amongst the output variables in structured prediction; however, their reliance on gradient based inference (GBI) makes the inference slow and unstable. In this work, we propose Structured Energy As Loss (SEAL) to take advantage of the expressivity of energy networks without incurring the high inference cost. This is a novel learning framework that uses an energy network as a trainable loss function (loss-net) to train a separate neural network (task-net), which is then used to perform inference through a forward pass. We establish SEAL as a general framework wherein various learning strategies like margin-based, regression, and noise-contrastive could be employed to learn the parameters of loss-net. Through extensive evaluation on multi-label classification, semantic role labeling, and image segmentation, we demonstrate that SEAL provides various useful design choices, is faster at inference than GBI, and leads to significant performance gains over the baselines.

## 1 Introduction

Structured prediction is an important machine learning problem in which, given an input $\mathbf{x}$, the model needs to predict the output $\mathbf{y} \in \mathcal{Y}_1 \times \mathcal{Y}_2 \times \cdots \times \mathcal{Y}_L$ where there are dependencies across $y_1, \ldots, y_L$. Common structured prediction tasks include multi-label classification (Belanger & McCallum, 2016; Gygli et al., 2017), extracting parse trees or semantic role labels from texts (Palmer et al., 2010), and image segmentation (Müller, 2014). Traditionally, these tasks are performed using models that explicitly represent the dependencies among output variables, such as directed and undirected graphical models, including Hidden Markov Models and Conditional Random Fields. However, except in cases with very limited dependency patterns (Lafferty et al., 2001; Ghamrawi & McCallum, 2005), inference algorithms in these methods are difficult to engineer and expensive to run because structured output spaces are exponentially large. In a multi-label classification task, for example, the size of the label space $\mathcal{Y}$ is $2^L$ given $\mathcal{Y}_i = \{0, 1\}$.

With their growing expressivity and capacity, it is common to use large feedforward neural networks to implicitly capture dependencies among output variables. Given an input $\mathbf{x}$, the elements of an output $\mathbf{y}$ are predicted in a conditionally independent manner in parallel, solely relying on the representation of input $\mathbf{x}$ to capture dependencies among output variables. While the feedforward approach is computationally efficient, the widely used cross-entropy loss does not directly capture

dependencies in the output space and thus often lacks statistical efficiency in capturing multivariate correlations.

A recent line of methods (Belanger & McCallum, 2016; Gygli et al., 2017; Rooshenas et al., 2019) model the joint space of $\mathbf{x}$ and multivariate $\mathbf{y}$ explicitly with an energy function $E_\Theta(\mathbf{x}, \mathbf{y})$ (LeCun et al., 2006), which aims to learn arbitrary global dependencies in the output space. They relax the combinatorial output space $\mathcal{Y}$ to $\tilde{\mathcal{Y}} = [0, 1]^L$ and apply gradient-based inference (GBI) directly on $\tilde{\mathbf{y}}$ to search for optima on the relaxed energy surface as $\arg\min_{\tilde{\mathbf{y}}} E(\mathbf{x}, \tilde{\mathbf{y}})$. These energy models are therefore often called structured *prediction* energy networks (SPENs).

SPENs show noticeable gains in predictive performance over feedforward models and traditional graphical models such as (Chen et al., 2015; Schwing & Urtasun, 2015). Due to the use of GBI, however, inference for SPENs remains relatively inefficient when compared to feedforward approaches (Tu & Gimpel, 2019). Moreover, models using GBI are often difficult to train and sensitive to sampling methods and numerous hyperparameters, including step size, number of iterations, and initialization for GBI. This raises a question: *Can energy networks be used in a way that is as expressive as SPENs, as efficient at inference as feedforward approaches, and also easy to train?* To this end, we propose using structured energy networks as parameterized loss functions for feedforward networks, instead of as prediction networks.

In this paper, we introduce the **S**tructured **E**nergy **A**s **L**oss (`SEAL`) framework which uses trainable SPENs (loss-net[1]) as loss functions to guide the training of other feedforward networks (task-net). The key idea is to provide the feedforward network with access to rich dependencies in the output space through a learned loss-net (§2.1 `SEAL-static`). We further propose to learn the loss-net in a dynamic fashion (§2.2 `SEAL-dynamic`) by adjusting the loss-net to be tailored to the most up-to-date outputs of the feedforward network; and then propose an NCE ranking loss uniquely suited for the `SEAL-dynamic` framework. Using feature-based multi-label classification as a case study, we extensively analyze the impact of various loss functions for updating the loss-net (§2.3). We further demonstrate the faster inference and higher performance of `SEAL`, compared to other baseline methods, over feature-, text-based multi-label classification, semantic role labeling, and image segmentation (§4). Additionally, we show that energy models capture the dependencies on the output space through analyzing gradient signals in an ablation study (§5 and Appendix G).

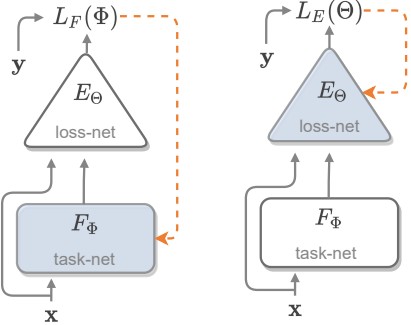

Figure 1: Overview of the `SEAL` framework. The figure on the left describes the update to the task-net in Eq.(3) and Eq.(5) and the figure on the right describes the update to the loss-net in Eq.(4).

---

Algorithm 1: SEAL-dynamic Algorithm

---

**Require:** $(\mathbf{x}, \mathbf{y})$: Training Instance
**Require:** $F_\Phi$: Feedforward Network
**Require:** `sampling`: True/False flag
**Require:** optimizer$_\Theta$, optimizer$_\Phi$
**Require:** $T$: No. of steps
   $\Theta_0, \Phi_0 \leftarrow$ Random initialization, $t \leftarrow 0$
   **while** $t < T$ **do**
      **if** `sampling` **then**
         $S \leftarrow \{\mathbf{y}^{(i)}, i = 1, \ldots, K | \mathbf{y}^{(i)} \sim F_{\Phi_t}(\mathbf{x})\}$
      **else**
         $S \leftarrow \{F_{\Phi_t}(\mathbf{x})\}$       ▷ singleton set
      $t \leftarrow t + 1$
   Update $\Theta$ as Eqn. 4 using $\tilde{\mathbf{y}} \in S$ and optimizer$_\Theta$
   Update $\Phi$ as Eqn. 5 using optimizer$_\Phi$

---

## 2 Structured Energy Network as Loss (`SEAL`)

**Notation.** $\mathcal{X}$ denotes the input space, $\mathcal{Y} = \{0, 1\}^L$ the label space, and $\tilde{\mathcal{Y}} = [0, 1]^L$ the continuous relaxation of $\mathcal{Y}$. Let $\mathcal{D} = \{(\mathbf{x}^{(j)}, \mathbf{y}^{(j)}) \in \mathcal{X} \times \mathcal{Y}\}_{j=1}^N$ denote the dataset, and $y_i \in \{0, 1\}$ the $i$-th dimension in the label vector $\mathbf{y}$ for $i \in \{1, \ldots, L\}$, and $\tilde{y_i} \in [0, 1]$ its continuous relaxation. The feedforward network $F_\Phi : \mathcal{X} \to \tilde{\mathcal{Y}}$ is a parametrized function that maps an input $\mathbf{x} \in \mathcal{X}$ to an output

---

[1]We refer an energy network as loss-net when it is used as a trainable loss function for another network.

$F_\Phi(\mathbf{x}) \in \tilde{\mathcal{Y}}$, where $F_\Phi(\mathbf{x})_i$ is the predicted probability of label $i$ occurring, given the input $\mathbf{x}$. The structured energy network $E_\Theta : \mathcal{X} \times \tilde{\mathcal{Y}} \to \mathbb{R}$ is a parameterized function that maps an input $\mathbf{x}$ and a continuously relaxed output $\tilde{\mathbf{y}}$ to real-valued scalar energy. For clarity, we refer to the feedforward network as *task-net*, and the structured energy network as *loss-net*.

As depicted in Figure 1, the proposed `SEAL` framework consists of two parametric components. First, the loss-net $E_\Theta$ is trained by an energy loss $L_E$ to capture an energy surface that reflects the latent dependencies among output variables. Low energy indicates a high quality of a label prediction given an input. Second, a task-net $F_\Phi$ is trained using a loss $L_F$ that involves minimizing the energy $E_\Theta$ estimated by the loss-net. After training finishes, the loss-net is no longer utilized; inference time predictions are obtained solely from the task-net as $\hat{y}_i = \mathbf{1}(F_\Phi(x)_i \geq 0.5)$. Unlike SPEN which requires iterative gradient-based inference, a task-net trained with the SEAL framework has the same computational efficiency at inference-time as standard feedforward networks (Appendix D), while enjoying a loss that captures the label dependencies through the loss-net.

To guide the task-net training with the loss-net $E_\Theta$, we propose the task-net training loss $L_F(\Phi; \Theta)$ to be a weighted sum of estimated energy and a binary-cross entropy (BCE) loss as

$$L_F(\Phi; \Theta) = \lambda_1 E_\Theta(\mathbf{x}, F_\Phi(\mathbf{x})) + \lambda_2 \sum_{j=1}^{L} \text{BCE}(y_j, F_\Phi(\mathbf{x})_j) \tag{1}$$

where $(\mathbf{x}, \mathbf{y})$ is a training instance and $\text{BCE}(y_j, F_\Phi(\mathbf{x})_j) = -[y_j \log F_\Phi(\mathbf{x})_j + (1 - y_j) \log (1 - F_\Phi(\mathbf{x})_j)]$. By incorporating the energy term in $L_F$, we enable the loss function to reflect the latent variable dependencies in the output space, whereas BCE treats $L$ labels in a conditionally independent fashion.

We present the formulation of $E_\Theta$ used for multi-label classification to analyze the effect of introducing $E_\Theta$ into $L_F$ without loss of generality. We use energy network $E_\Theta(x, y)$ that has the same structure with Belanger & McCallum (2016) and Gygli et al. (2017) for a fair comparison with them. The energy $E_\Theta$ is the sum of local energy $E_\Theta^{\text{local}}$ and global energy $E_\Theta^{\text{global}}$ which are defined as:

$$E_\Theta^{\text{local}}(\mathbf{x}, \mathbf{y}) = \sum_{i=1}^{L} y_i \mathbf{b}_i^\top T_E(\mathbf{x}), \ E_\Theta^{\text{global}}(\mathbf{x}, \mathbf{y}) = \mathbf{v}^\top \gamma(\mathbf{M}\mathbf{y}), \tag{2}$$

where $\mathbf{b}_i, \mathbf{v}, \mathbf{M}, T_E$ contain learnable parameters, $T_E$ is a feature network to represent an input $\mathbf{x}$ in the loss-net, and $\gamma(z) = \log(1 + e^z)$ is the softplus activation function. We observe from the derivative of the global energy $\frac{\partial E_\Theta^{\text{global}}}{\partial y_i} = \left\{ M^T \text{Diag}(\gamma'(M\mathbf{y}))\mathbf{v} \right\}_i$ that all dimensions of $\mathbf{y}$ decides the gradient signal for each label $i$. The global energy term $E_\Theta^{\text{global}}$ in $L_F$, specifically, allows the gradient of the task-net loss to capture the dependency of $y_i$ on all $L$ dimensions of $\mathbf{y}$. In contrast, the gradient of BCE for each label $i$ only depends on the $i$-th dimension of $\mathbf{y}$. More details are in Appendix A.

## 2.1 `SEAL`-static

Since the task-net loss $L_F$ depends on the quality of the loss-net, it is critical to find the loss-net parameters $\Theta$ that produce the best loss surface for training the task-net $\Phi$. In order to achieve this, we first estimate the loss-net parameters $\widehat{\Theta}$ over training data and then optimize the task-net parameters $\Phi$ by plugging in the fixed loss-net parameters $\widehat{\Theta}$ into $L_F$. Formally, given training instances $(\mathbf{x}, \mathbf{y}) \in \mathcal{D}$, the task-net parameters $\Phi$ are trained in the following manner:

$$\min_\Phi \frac{1}{|\mathcal{D}|} \sum_\mathcal{D} L_F(\Phi; \widehat{\Theta}) \qquad \text{s.t. } \widehat{\Theta} = \arg\min_\Theta \frac{1}{|\mathcal{D}|} \sum_\mathcal{D} L_E(\mathbf{x}, \mathbf{y}, \tilde{\mathbf{y}}; \Theta) \tag{3}$$

Since the loss function $L_F$ is static after pretraining $\widehat{\Theta}$, we call this approach `SEAL`-static. We discuss various types of $L_E$ losses in §2.3.

## 2.2 `SEAL`-dynamic

Ideally, we would like to learn a perfect energy surface by estimating $\widehat{\Theta}$ that is static. However, estimating an accurate energy surface over a high-dimensional, continuous joint space $\mathcal{X} \times \tilde{\mathcal{Y}}$ is a

challenging task. Instead, we hypothesize that using loss-net to dynamically estimate the energy surface that is focused around the current input and task-net output pair $(\mathbf{x}, F_\Phi(x))$ could be more beneficial in teaching task-net. To this end, we propose SEAL-dynamic which dynamically adapts loss-net so that the region of interest for training task-net is well represented.

Given a batch of training data $B_t$, a complete training step of SEAL-dynamic is provided as

$$\Theta_t \quad \leftarrow \quad \Theta_{t-1} - \nabla_\Theta \frac{1}{|B_t|} \sum_{B_t} \mathrm{L}_E \left( \mathbf{x}, \mathbf{y}, F_{\Phi_{t-1}}(\mathbf{x}); \Theta \right) \tag{4}$$

$$\Phi_t \quad \leftarrow \quad \Phi_{t-1} - \nabla_\Phi \frac{1}{|B_t|} \sum_{B_t} \mathrm{L}_F(\Phi; \Theta_t) \tag{5}$$

where the loss-net parameters $\Theta_t$ and the task-net parameters $\Phi_t$ are optimized in alternating fashion. Note that, from Eq.(3) to Eq.(4), arbitrary label samples $\tilde{\mathbf{y}}$ were replaced with $F_{\Phi_{t-1}}(\mathbf{x})$ from the task-net during the training of $\Theta_t$ parameters. Thus the energy surface learned by the loss-net is adaptive to the task-net. On the contrary, in SEAL-static, the training of the static loss-net is independent of the task-net. Further, the update step for the task-net parameters $\Phi_t$ relies on dynamic loss-net parameters $\Theta_t$ rather than static $\widehat{\Theta}$. In §4, we show the effectiveness of the proposed SEAL framework compared to standard feedforward networks and SPENs, and further that SEAL-dynamic often improves over SEAL-static.

## 2.3 Energy losses $\mathbf{L}_E$

We now discuss various energy loss functions $\mathrm{L}_E$ with different characteristics such as margin-based, regression-based, and noise-contrastive-sampling-based, that can be plugged into the SEAL framework to train the loss-net. For brevity, we use $\mathrm{L}_E(\Theta)$ and $\mathrm{L}_E(\mathbf{x}, \mathbf{y}, \tilde{\mathbf{y}}; \Theta)$ interchangeably.

**Margin-based ($\mathbf{L}_{E-\mathbf{margin}}$).** SPEN (Belanger & McCallum, 2016) learns structured energy network with the SSVM loss (Taskar et al., 2004; Tsochantaridis et al., 2004) so that energy function learns to predict sufficient energy difference, larger than the margin $\Delta(\tilde{\mathbf{y}}, \mathbf{y})$, between an arbitrary output $\tilde{\mathbf{y}}$ and a true output $\mathbf{y}$. To examine the effect of the margin-based loss in SEAL, we follow SPEN and utilize the SSVM loss as

$$\mathrm{L}_{E-\mathrm{margin}} = \sum_{\mathbf{x},\mathbf{y}} \max_{\tilde{\mathbf{y}}} \left[ \Delta(\tilde{\mathbf{y}}, \mathbf{y}) - E_\Theta(\mathbf{x}, \tilde{\mathbf{y}}) + E_\Theta(\mathbf{x}, \mathbf{y}) \right]_+. \tag{6}$$

SPEN resorts to gradient ascent on $\tilde{\mathbf{y}}$ to solve the optimization problem $\arg\max_{\tilde{\mathbf{y}}} \Delta(\tilde{\mathbf{y}}, \mathbf{y}) - E_\Theta(\mathbf{x}, \tilde{\mathbf{y}})$. InfNet (Tu & Gimpel, 2018) proposes an adversarial training of the task-net $\Phi$ together with the energy network $\Theta$. We identify InfNet as a special case of SEAL-dynamic that uses $\mathrm{L}_{E-\mathrm{margin}}$. We defer a detailed description of InfNet to the related work section §3.

**Regression-based ($\mathbf{L}_{E-\mathbf{regression}}$).** Deep Value Network (DVN) (Gygli et al., 2017) attempts to learn an energy network which directly outputs a score that is similar to the metric $s(\tilde{\mathbf{y}}, \mathbf{y})$ of interest, such as F1 score, that compares arbitrary $\tilde{\mathbf{y}}$ with true output $\mathbf{y}$. Following DVN, we let the true score $s(\cdot)$[2] and $\sigma(-E(\cdot))$ be between $[0, 1]$ and express the regression-based loss as cross entropy between $s$ and $\sigma(-E)$:

$$\mathrm{L}_{E-\mathrm{regression}} \quad = \quad -s(\tilde{\mathbf{y}}, \mathbf{y}) \log \sigma(-E_\Theta(\mathbf{x}, \tilde{\mathbf{y}})) - (1 - s(\tilde{\mathbf{y}}, \mathbf{y})) \log(1 + \sigma(-E_\Theta(\mathbf{x}, \tilde{\mathbf{y}}))) \tag{7}$$

where $\sigma(\cdot)$ is a sigmoid function. DVN utilizes two samples to train the model: the model prediction that minimizes the energy and an adversarial sample $\tilde{\mathbf{y}}$ that maximizes the loss $\mathrm{L}_{E-\mathrm{regression}}$. We follow this approach for training SEAL-static with $\mathrm{L}_{E-\mathrm{regression}}$. To train SEAL-dynamic with $\mathrm{L}_{E-\mathrm{regression}}$, we simply use the output of task-net, $F_{\Phi_{t-1}}(\mathbf{x})$, as described in Eq.(4).

To better capture the energy surface in SEAL-dynamic, we further introduce *regression-s* loss $\mathbf{L}_{E-\mathbf{regression\text{-}s}} = \sum_{\tilde{\mathbf{y}} \in S} \mathrm{L}_{E-\mathrm{regression}}(\mathbf{x}, \mathbf{y}^*, \tilde{\mathbf{y}}; \Theta)$ where the sample set $S$ is taken by continuously perturbing task-net output $F_{\Phi_{t-1}}(\mathbf{x})$ with Gaussian noise (experimental analysis on the 'effect of samples' can be found at Appendix E.1).

---

[2]In this paper, we adopt soft F1 score $s(\tilde{\mathbf{y}}, \mathbf{y})$ from Gygli et al. (2017) that is defined on the continuous $\tilde{\mathbf{y}} \in [0, 1]^L$.

**Noise-contrastive ranking ($L_{E-\textbf{NCE}}$).** Lastly, we propose to use noise contrastive estimation (NCE) ranking loss (Ma & Collins, 2018) uniquely for SEAL-dynamic. We are interested in capturing appropriate energy surface in high-probability output spaces of task-net. To reflect such space accurately, we propose to train the loss-net $E_\Theta$ to contrast the true output $\mathbf{y}$ with a group of $K$ negative samples drawn from the task-net prediction $F_\Phi(\mathbf{x})$ motivated by NCE.

Before we discuss $L_{E-\text{NCE}}$ for SEAL-dynamic in more detail, we first review the original form of NCE ranking loss from Ma & Collins (2018). For $K$ samples $\mathbf{y}^{(k)} \sim P_N$, $k = 1, \ldots, K$, drawn from a noise distribution $P_N$, rewriting ground truth $\mathbf{y}$ as $\mathbf{y}^{(0)}$ without loss of generality, the NCE ranking loss is defined as

$$L_{E-\text{NCE}} = -\log \frac{\exp\left(s(\mathbf{x}, \mathbf{y}^{(0)}; \Theta)\right)}{\sum_{k=1}^{K} \exp\left(s(\mathbf{x}, \mathbf{y}^{(k)}; \Theta)\right)} \text{ where } s(\mathbf{x}, \mathbf{y}; \Theta) = -E_\Theta(\mathbf{x}, \mathbf{y}) - \log P_N(\mathbf{y}). \quad (8)$$

Given the noise samples $y^{(k)}$, minimizing Eq.(8) minimizes the energy of the ground truth sample $\mathbf{y}^{(0)}$ while increasing the energy of the noise samples. In SEAL-dynamic, we propose to use $\prod_i P(y_i|\mathbf{x}; \Phi_t)$, which is dynamically changing, in place of the static $P_N(\mathbf{y})$ in Eq.(8). The NCE ranking loss is thus

$$\text{Eq.(8) but with } s(\mathbf{x}, \mathbf{y}; \Theta) = -E_\Theta(\mathbf{x}, \mathbf{y}) - \sum_i \log P(y_i|\mathbf{x}; \Phi_t). \quad (9)$$

The novelty of this proposal is that task-net, viewed as producing a dynamic noise distribution, is in turn trained using loss-net.

The NCE ranking loss $L_{E-\text{NCE}}$ brings two benefits in SEAL-dynamic. First, $L_{E-\text{NCE}}$ captures the output distribution of the task-net rather than just a single point. Second, $L_{E-\text{NCE}}$ allows estimation of a more fine-grained loss surface as the training steps proceed. In our SEAL-dynamic learning procedure, we hypothesize that $\prod_i P(y_i|\mathbf{x}; \Phi_t)$ becomes closer to the true data distribution as training steps proceed. Contrasting these noise samples to the ground truth could lead the loss surface estimated by loss-net to be more fine-grained. This is in line with the known fact that NCE methods work the best when the noise distribution is close to the data distribution but not the same (Gutmann & Hirayama, 2012).

To thoroughly experiment with the proposed NCE ranking loss $L_{E-\text{NCE}}$, we also train a static energy network with $L_{E-\text{NCE}}$ in order to utilize it as a prediction network with GBI and as a loss-net in SEAL-static. We set $P_N(\mathbf{y}) = \prod_i P(y_i|\mathbf{x}; \Psi)$ in Eq.(8), where $\Psi$ is a fixed pretrained feedforward network. When we train an energy network $E_\Theta$ with $L_{E-\text{NCE}}$, $\frac{\exp(-E_\Theta(\mathbf{x},\mathbf{y}))}{\sum_{\mathbf{y} \in \mathcal{Y}} \exp(-E_\Theta(\mathbf{x},\mathbf{y}))}$ becomes an unbiased estimator of the true distribution $P(\mathbf{y}|\mathbf{x})$ (Ma & Collins, 2018). Given a perfect energy $E_\Theta(\mathbf{x}, \mathbf{y})$, minimizing it with respect to $\mathbf{y}$ in SEAL-static and for prediction with GBI is equivalent to minimizing the estimate of $-\log P(\mathbf{y}|\mathbf{x})$.

## 3 Related Work

**Learning with dynamic loss.** Wu et al. (2018) and Huang et al. (2019) attempt to learn a dynamic loss for multi-class classification tasks. Both approaches try to learn a parametrized loss function that can directly increase the model's metric score $m$, such as the BLEU or 0-1 accuracy, on the validation set. Since $m$ is usually non-differentiable, Huang et al. (2019) view this problem as a reinforcement learning problem and attempt to learn a loss function that maximizes the expected reward. On the other hand, Wu et al. (2018) relax the $m$ to be differentiable and try to learn a loss parameter ($\Theta$) that can improve the relaxed score on feed-forward ($\Phi$) output.

There are three major differences between SEAL and the previous work. While both Wu et al. (2018); Huang et al. (2019) use the parametric form $\sigma(\mathbf{y}^T \Theta \log F_\Phi(x))$ to define the loss function, SEAL utilizes a bigger class of functions expressed using general energy network $E_\Theta$. Moreover, SEAL does not require manually designed state vectors and works directly with the task-net outputs. Lastly, while the SEAL framework is designed for structured prediction, Wu et al. (2018) and Huang et al. (2019) are designed for multi-class classification and it is non-trivial to extend these methods for structured prediction.

**Structured Prediction Energy Network (SPEN).** Structured prediction energy networks (Belanger & McCallum, 2016) and its variants (Gygli et al., 2017; Rooshenas et al., 2019) (together referred to as SPENs) learn an energy network $E_\Theta : \mathcal{X} \times \mathcal{Y} \to \mathbb{R}$ and predict the output using gradient-based inference (GBI) with the objective $\mathbf{y} = \arg\min_y E(\mathbf{x}, \mathbf{y})$. The optimization objective for training the SPENs relies on finding adversarial or margin-violating points in the label space using gradient based inference (GBI). This learning process in SPENs is highly sensitive with respect to hyper-parameters such as the learning rate and initial points for the GBI, resulting in high variance in the model performance. Nonetheless, SPENs are strong baselines for structured prediction tasks and are used as motivations for the design of SEAL. They are further discussed in §2.3.

**Jointly learning $\Theta$ and $\Phi$.** Inference Network (InfNet) (Tu & Gimpel, 2018) is the first work that interactively learns a structured energy network and a feedforward network. It aims to replace the expensive GBI in SPEN with a feedforward network's forward pass. During its training with the SSVM loss (Eq.(6)), SPEN maximizes $\Delta(\tilde{\mathbf{y}}, \mathbf{y}) - E_\Theta(\mathbf{x}, \tilde{\mathbf{y}})$ with respect to $\tilde{\mathbf{y}}$ by GBI. InfNet, on the contrary, trains a feedforward network $F_\Phi$ to generate $\tilde{\mathbf{y}} = F_\Phi(x)$ that maximizes the term. This results in an adversarial framework (Goodfellow et al., 2014), where a feedforward network maximizes the margin-based loss while an energy network minimizes it. The approach, however, suffers from train-test-time objective mismatch (Tu et al., 2020). While the feedforward network serves as an adversarial sampler during training, the end goal is to learn a feedforward network that predicts outputs with low energy at test time. Tu et al. (2020) resolves this by learning two separate feed-forward networks: one adversarial sampler for training and the other for performing inference.

Our proposed SEAL generalizes, and differs from, InfNet by providing a framework that can employ completely independent loss functions for training loss-net and task-net. SEAL updates task-net in the direction of reducing energy defined by loss-net; loss-net is only used as a tool for evaluation of loss, and how loss-net is trained does not matter as long as it estimates a good loss surface for task-net. The SEAL framework is not interested in producing the similar test-time behaviour of SPEN as InfNet did. Despite the difference in the end goal, mechanically speaking, InfNet is a special case of SEAL-dynamic (i.e. SEAL-dynamic with $\mathrm{L}_{E-\mathrm{margin}}$) with the margin removed from the SSVM loss for the feedforward model training. In fact, Tu et al. (2020) reports that this special case leads to more efficient and stable learning of the feedforward network . We use this special case of InfNet to represent SEAL with $\mathrm{L}_{E-\mathrm{margin}}$ in the main experiment, and by doing so, we reinterpret the success of InfNet as a result of utilizing energy network as a loss function, as our formulation shows, rather than the result of task-net mimicking the test-time behaviour of energy network.

## 4 Experiments

**Tasks and Datasets.** We demonstrate the effectiveness of SEAL on widely deployed structured prediction tasks: feature and text-based multi-label classification (MLC), semantic role labeling (SRL), and image segmentation. Feature-based MLC is a standard evaluation task in previous works on SPENs (Belanger & McCallum, 2016; Gygli et al., 2017; Tu & Gimpel, 2018). We utilize 7 feature-based MLC datasets which cover various label sizes (ranging from 27 to 4k), training sizes, and input characteristics (binary, real) as shown in Table 5 to extensively compare baselines and proposed SEAL approaches: cross-entropy (CE), energy network, SEAL-static, and SEAL-dynamic. Then we verify that SEAL-dynamic, which outperforms SEAL-static on feature-based MLC, can also effectively train large pre-trained language models like BERT (Devlin et al., 2019) on text-based MLC and SRL tasks and convolutional networks on image segmentation. Dataset statistics are described in Appendix B.

**Feature network, loss-net and task-net.** In MLC and SRL experiments, to represent input $\mathbf{x}$, we utilize the same feature network structures $T_E$ and $T_F$ for the energy and feedforward networks, which correspond to the loss-net and the task-net for SEAL framework. The energy network utilizes $T_E$ in the manner shown in equation 2. Given a feature network $T_F$ that generates features in $\mathbb{R}^h$, the task-net $F_\Phi(x) = \mathbf{G}T_F(x)$, where $\mathbf{G} \in \mathbb{R}^{L \times h}$ is a matrix consisting of learnable embeddings, one row of embedding for each label. Specific feature network structures vary by tasks: multi-layer perceptron for feature-based MLC and BERT for text-based MLC and SRL. In image segmentation experiments, we follow Gygli et al. (2017) and use a Fully Convolutional Network (FCN) as the task-net; the loss-net uses the same convolutional layers in the FCN, followed by fully connected

Table 1: Performance (F1 ↑) of feature-based MLC datasets. The task-net learned with SEAL-static and SEAL-dynamic outperforms cross-entropy-trained model most of the time. The SEAL framework which utilizes an energy network as a loss performs better and faster than an energy network utilized as a prediction network. Best results per dataset are marked with underline; best results within a framework (energy network with GBI, SEAL-static, SEAL-dynamic) are marked in **bold**.

| method | samples | discrete input datasets | | | cal500 | continuous input datasets | | | mean F1 gain over CE | |
| | | bibtex | delicious | genbase | | eurlexev | expr_fun | spo_fun | overall | −genbase |
|---|---|---|---|---|---|---|---|---|---|---|
| cross-entropy (CE) | x | 42.40 ± 0.18 | 29.89 ± 0.08 | 47.37 ± 7.47 | 33.58 ± 0.43 | 42.19 ± 0.07 | 37.50 ± 0.18 | 27.81 ± 0.13 | - | - |
| **energy network with GBI** | | | | | | | | | | |
| SPEN | x | 42.99 ± 0.16 | 24.20 ± 0.07 | 32.13 ± 0.15 | 37.24 ± 1.07 | **41.86 ± 0.04** | **36.74 ± 0.61** | 27.95 ± 0.18 | -2.52 | -0.40 |
| DVN | x | **45.95 ± 0.13** | **24.87 ± 0.57** | **77.92 ± 1.00** | **47.74 ± 0.19** | 25.49 ± 0.32 | 31.47 ± 0.43 | **29.35 ± 0.31** | 3.15 | -1.42 |
| NCE ranking | o | 12.95 ± 0.53 | 12.69 ± 0.15 | 12.40 ± 0.81 | 33.89 ± 0.49 | 0.19 ± 0.00 | 27.74 ± 0.17 | 18.06 ± 0.38 | -20.40 | -17.98 |
| **SEAL-static** | | | | | | | | | | |
| margin | x | **43.11 ± 0.15** | 28.08 ± 0.06 | 57.45 ± 5.17 | 33.91 ± 0.45 | 42.15 ± 0.08 | **38.13 ± 0.17** | 28.15 ± 0.17 | 1.46 | 0.02 |
| regression | x | 42.29 ± 0.18 | **30.09 ± 0.08** | **96.68 ± 0.11** | 37.63 ± 0.44 | 42.18 ± 0.07 | 38.12 ± 0.17 | 28.42 ± 0.09 | 7.78 | 0.86 |
| NCE ranking | o | 43.03 ± 0.15 | 30.08 ± 0.05 | 96.43 ± 0.15 | **37.82 ± 0.38** | **42.11 ± 0.11** | 37.78 ± 0.16 | 28.29 ± 0.05 | 7.83 | 0.95 |
| **SEAL-dynamic** | | | | | | | | | | |
| margin (InfNet) | x | 42.86 ± 0.16 | 29.75 ± 0.11 | 96.53 ± 0.21 | 36.69 ± 0.40 | 41.83 ± 0.65 | 37.81 ± 0.14 | 28.43 ± 0.13 | 7.59 | 0.67 |
| regression | x | 43.74 ± 0.14 | 29.79 ± 0.05 | 96.95 ± 0.14 | 37.97 ± 0.33 | 41.67 ± 0.08 | 37.99 ± 0.09 | **29.02 ± 0.11** | 8.05 | 1.13 |
| regression-s | o | 44.53 ± 0.18 | 29.87 ± 0.07 | 96.81 ± 0.20 | 38.95 ± 0.13 | 42.37 ± 0.05 | 37.93 ± 0.12 | 28.29 ± 0.12 | 8.29 | 1.43 |
| NCE ranking | o | **44.76 ± 0.17** | **34.67 ± 0.38** | **97.32 ± 0.81** | 41.62 ± 0.80 | **42.77 ± 0.06** | **38.28 ± 0.09** | 28.83 ± 0.13 | **9.64** | **2.93** |
| **Inference speed** | feedforward | 840 | 2307 | 1005 | 1080 | 317 | 3801 | 5231 | Average | 3.63x |
| (examples/sec) | GBI | 414 | 599 | 574 | 438 | 162 | 638 | 709 | speedup | |

layers which replaces the tranposed convolutional layers in FCN to output scalar scores, instead of the energy formulation in Eq.(2).

**Training.** We use separate ADAM optimizers (Kingma & Ba, 2014) for the loss-net (Θ) and the task-net (Φ) which optimize parameters in alternating fashion[3]. We defer specific training details such as hyperparameters, gpu environment, and number of random seed runs to the Appendix C.1.

### 4.1  Multi-label classification for feature-based datasets

Multi-label classification (MLC) is a structured prediction task that does not hold an obvious latent structure in the multivariate space. In addition to including Bibtex and Delicious that were used in previous works (Belanger & McCallum, 2016; Gygli et al., 2017; Tu & Gimpel, 2018), we carefully picked smaller (Genbase, Cal500) and larger datasets (Eurlex-ev) as well as datasets with known taxonomy (Spo_FUN, Expr_FUN) to examine SEAL in various scenarios. The experiment results (test F1) across 7 feature-based dataset are presented in Table 1: feedforward network trained with cross-entropy (row 1), energy networks (row 2-4) described in §3 that is evaluated with GBI, task-nets trained with SEAL-static (row 5-7), and task-nets trained with SEAL-dynamic (row 8-11).

In general, SEAL-static and SEAL-dynamic outperform both cross-entropy (CE) and prediction energy networks with GBI on almost all datasets with average gains of SEAL-dynamic over CE ranging from +0.67 to +2.93 F1 points, excluding results on genbase dataset. While the near-perfect performance of SEAL on genbase is interesting (see Appendix E.2 for detailed analysis), we exclude this excessive gain for computing the average performance when comparing different models throughout this paper.

Since SEAL uses single forward pass of the task-net for inference, it is as fast as the feedforward model and 3.63x faster (on average) at inference than the corresponding energy network trained using GBI. However, in training time, SEAL utilizes about twice the parameter to feedforward model as it loads both task-net and loss-net and takes about 5x more time to train compared to feedforward model. In summary, SEAL is better performing than both energy network and feedforward model and as fast as feedforward model at inference time by utilizing more computations in training time. The training time, inference time, and parameter size of different methods are reported in Appendix D.

**Cross entropy vs SEAL.**  While cross entropy cannot capture the dependencies between the labels, we find that global energy of the loss-net visibly captures some of these dependencies. We plot the gradients of global energy with respect to a pair of labels to analyse pairwise dependencies. These plots can be found in §5 and the Appendix G.

---

[3]The code we used to train and evaluate our models is available a `https://github.com/iesl/seal-neurips-2022`

**Energy network vs. `SEAL`-static.**   There is a high variability in the performance of the energy network with GBI. For instance, while SPEN and DVN surpass CE models with large margins for half of the datasets, they underperform on the rest of the datasets with similar margins. Further, even though DVN has the best performance on three datasets, its average performance (-genbase) is lower than CE. Nonetheless, in `SEAL`-static, when the identical energy network is utilized as a loss-net, the performances increase in a stable manner (0.42-2.26 F1 on average) with faster (1.8x-7.4x) inference time. This trend is most evident with the NCE ranking loss, where the energy network trained with it drastically fails to provide a good energy surface for predicting output with GBI, whereas it provides a useful loss surface for training task-net.

We believe there are two factors for this stable improvement of `SEAL`-static. One is that injection of an energy surface as a loss is a softer injection of gradients compared to it being used as a surface for inference. The second factor is a synergy between cross-entropy and the energy surface. Even when certain regions of the energy surface are less accurate, the cross-entropy signal could compensate for the erroneous surface. Thus, we claim that an energy network as a loss is a more stable way of utilizing it compared to GBI.

**`SEAL`-static vs. `SEAL`-dynamic.**   As seen in Table 1, with an average gain of 0.47 to 1.97 F1 points, `SEAL`-dynamic is always on par or better than the corresponding `SEAL`-static. This confirms our hypothesis that it is helpful for the loss surface to consider the current state of the task-net as opposed to estimating the global energy landscape. Lastly, `SEAL`-dynamic not only brings better performance but also simplifies the engineering as both loss-net and task-net can be trained from scratch. We observed that binary cross entropy in $L_F$ stabilizes the learning signal even in the early stage when loss-net might not be well-trained.

In summary, the `SEAL` framework outperforms CE loss and energy networks with GBI. We observe the generality of the `SEAL` framework as all of the energy loss types benefit working with `SEAL`, although more effective types of energy loss exist. `SEAL`-dynamic brings extra gain in performance to `SEAL`-static and among them, the NCE ranking loss shows the most notable improvement. Lastly, as can be seen with the example of the NCE ranking loss, an energy network that serves as a poor prediction network does not necessarily correlate to a poor energy loss for `SEAL`.

### 4.2   Multi-label classification for large text datasets

To test the effectiveness of `SEAL` on large language models, we finetune pre-trained BERT as the feature network $T_E$ and $T_F$ on text MLC datasets. Considering the computational expense of running hyper-parameter searches on BERT, we compare the best-performing `SEAL` from Table 1, `SEAL`-dynamic with NCE ranking, with baselines: BERT finetuned with cross-entropy and the state-of-the-art text-MLC model (LACO) (Zhang et al., 2021). Following LACO, we experiment with the Arxiv Academic Paper Dataset (AAPD) (Yang et al., 2018) which consists

Table 2: Test F1 for AAPD

| method | **F1** ↑ |
|---|---|
| BERT (cross-entropy) | 73.97 |
| LACO (Zhang et al., 2021) | 74.90 |
| SEAL-dynamic (NCE) | **74.95**±**0.31** |

of computer science papers with annotations of 54 topics. Table 2 shows that `SEAL`-dynamic with the NCE ranking loss is effective on pre-trained models that utilizes large text datasets as well. `SEAL`-dynamic again improves task-net trained with cross entropy and performs on par with MLC-specific model architecture. To efficiently work with large pre-trained models, we further studied utilizing BERT-adapter models on three additional text-based datasets (Appendix F) which also showed `SEAL`-dynamic-NCE model outperforming cross-entropy and InfNet.

### 4.3   Semantic role labeling

To examine `SEAL` on capturing label dependencies at a sequence level, we perform experiments on semantic role labeling (SRL) (Palmer et al., 2010) using standard benchmark (CoNLL-12) (Pradhan et al., 2013). SRL aims to annotate the predicate-argument structure on plain text extracting *who* did *what* to *whom*. Because the argument roles in a sentence are dependent to each other, injection of structural priors has shown improvements in performance (Lee et al., 2019; Mehta et al., 2020; Li et al., 2020). To apply `SEAL`, we utilize pre-trained BERT as the feature network $T_E$ and $T_F$. To capture the energy on variable-length output, we modify $E_\Theta^{\text{global}}$ in equation 2 to use self-attention layers. We defer more detailed task description, data statistics, and energy architecture to appendix.

We compare SEAL-dynamic (NCE) on BERT-base against two models: one model trained with cross-entropy loss and another model trained with structured tuning framework (Li et al., 2020) where both models utilize BERT-base with CRF layers (Lafferty et al., 2001). Structured tuning enforces the model to learn the SRL-specific constraint (unique core role) through differentiable constraint loss. Unique core role violation refers to model predicting a type of core argument more than once when it should be unique. To examine effectiveness of SEAL in capturing output dependencies, we report unique-core-role violation rate (UCR) in percentage along with F1 score. While SEAL-dynamic does not enforce any specific constraints nor utilize CRF layers, it shows notable reduction in constraint violation and higher improvement in performance.

Table 3: Results on CoNLL-12. We report F1 and unique core role violation (UCR) in percentage.

| | **Dev** | | **Test** | |
| method | F1 ↑ | UCR ↓ | F1 ↑ | UCR ↓ |
|---|---|---|---|---|
| BERT-CRF | 85.92 | 1.41 | $85.97_{\pm0.14}$ | 1.32 |
| Structured tuning | 86.18 | **0.78** | $86.12_{\pm0.06}$ | **0.79** |
| SEAL-dynamic | **86.94** $_{\pm\,0.07}$ | $0.92_{\pm0.02}$ | $\mathbf{86.90_{\pm0.06}}$ | $0.91_{\pm0.03}$ |

Table 4: Test mean image IoU for Weizmann Horse segmentation.

| method | Mean IoU ↑ |
|---|---|
| FCN (cross-entropy) | $77.80_{\pm0.59}$ |
| DVN with GBI | $72.93_{\pm0.52}$ |
| SEAL-dynamic (regression) | $\mathbf{79.23_{\pm0.35}}$ |
| SEAL-dynamic (NCE) | $76.78_{\pm0.60}$ |

## 4.4 Image segmentation

We evaluate SEAL on binary image segmentation using the Weizmann Horse Image dataset (Borenstein & Ullman, 2004). This task requires a model to learn to predict detailed structures of horse shapes from a scarce 160 training images of a low $32\times32$ resolution. Following Gygli et al. (2017), we implement Fully Convolutional Network (FCN) (Long et al., 2015) as our feedforward baseline and DVN (Gygli et al., 2017) as our energy network baseline. Given a 3-channel $L$-pixel RGB image input $\mathbf{x}$, FCN outputs the probability $\tilde{\mathbf{y}} \in [0,1]^L$ that each pixel belongs to a horse object; a binary segmentation mask can be obtained through $\mathbf{1}(\tilde{\mathbf{y}} \geq 0.5)$. In contrast, DVN takes a pair of image $\mathbf{x}$ and soft segmentation mask $\tilde{\mathbf{y}}$ to output their compatibility score and resort to gradient-based inference for prediction. Following Gygli et al. (2017), we train DVN by plugging the soft IoU score

$$s(\tilde{\mathbf{y}}, \mathbf{y}^*) = \frac{\sum_{i=1}^{M} \min(\tilde{y}_i, y_i^*)}{\sum_{i=1}^{M} \max(\tilde{y}_i, y_i^*)}, \quad \text{while IoU}(\mathbf{y}, \mathbf{y}^*) = \frac{\mathbf{y} \cap \mathbf{y}^*}{\mathbf{y} \cup \mathbf{y}^*} \text{ for } \mathbf{y}, \mathbf{y}^* \in \{0,1\}^L, \quad (10)$$

into the Eq.(7). For training with the SEAL framework, we adopt the FCN and DVN architectures for the task-net and the loss-net respectively. In our experiment, we examine SEAL-dynamic with regression which uses the same energy loss as DVN and SEAL-dynamic with the NCE loss because it performs the best in multiple other experiments. More dataset and model details are in Appendix B; experiment details are in Appendix C.2.

Table 4 shows test performances of FCN trained by cross-entropy, DVN, and FCN trained with SEAL-dynamic. Similar to §4.1, we observe that the GBI procedure in DVN training and inference is slow and unstable (Appendix B.3 for more details). While DVN does not outperform feedforward model learned with cross-entropy, the same energy network structure used as a loss-net in SEAL-dynamic (regression) improves the mean IoU of FCN by 1.43 IoU point and has less variance across random-seed runs. Additionally, we find that SEAL-dynamic with the NCE loss proposed for MLC does not improve FCN. This is likely because many negative samples in NCE are non-realistic as we sample each pixel independently and do not take into account the prior that nearby pixels on image are likely to be in the same class. We leave the exploration of other energy network architectures and sampling strategies in the image domain as future work.

## 5 Further analysis

In this section, we analyze (1) whether gradient of loss-net rightly captures multivariate dependencies, (2) the effect of ranking loss, and (3) we provide pointers to the data-specific analysis and discussion on the effect of samples in Appendix E.

**Visualization of gradients** In order to inspect the learning signal provided to the task-net by the loss-net, we plot a single component of the gradient from the global energy of the loss-net, i.e.

$\frac{\partial E_\Theta^{\text{global}}}{\partial y_k}(y_i, y_k)$. In Figure 2, we inspect the global energy function of DVN trained on the expr_fun, a dataset that provides a taxonomy on the output space. We first plot the discussed gradient component when $l_k$ have positive association with $l_i$, i.e., $l_i \rightarrow l_k$[4] (2b). Contrasting this with the case when $l_k$ does not have a significant association with $l_i$ (2c), we see that the gradient of the global energy rightly captures the presence and absence of association between the labels. In contrast, binary cross-entropy is incapable of capturing label-to-label association (2a) which is obvious when we inspect its gradient as shown in Appendix A. See Appendix G for more examples and for detailed description of plotting procedure.

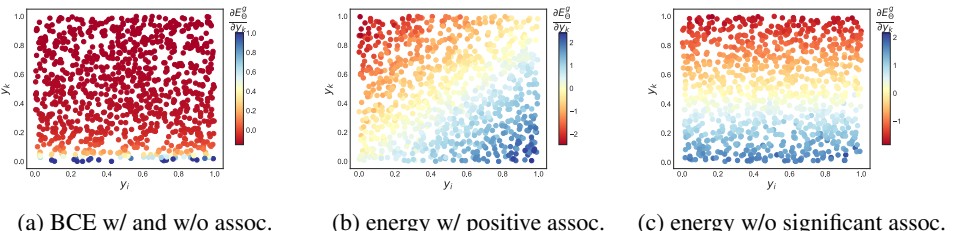

| (a) BCE w/ and w/o assoc. | (b) energy w/ positive assoc. | (c) energy w/o significant assoc. |

Figure 2: Given there exists a positive association $l_i \rightarrow l_k$ between two labels, the gradient $\frac{\partial E_\theta^{\text{global}}}{\partial y_k}$ is plotted (b) and compared with the gradient of BCE (a) when true $l_k = 1$. The gradient of global energy is also plotted (c) for the case when there is no significant association between the two labels.

**Effect of applying a ranking loss directly on $F_\Phi$**   With the large improvement that SEAL-NCE achieves, we examine whether the gain is coming from our SEAL framework or simply from the power of a ranking loss itself that can also be applied to the task-net $F_\Phi$ directly (i.e. $s(\mathbf{x}, \mathbf{y}; \Theta) = \sum_i \log P(\mathbf{y}_i|\mathbf{x}; \Phi_t)$ in equation 8). We conduct this ablation study over relatively small datasets: genbase, cal500, and delicious. While we observe a near +10 F1 point in genbase, we see a decrease in performance on cal500 (-1.0 F1) and delicious (-2.3 F1) compared to the plain cross-entropy model. With these experiments, we first reaffirm that genbase is an outlier dataset where capturing structure can significantly increase the performance. Second, we conclude that the ranking loss without an loss-net is not very effective and that the loss-net indeed plays a major role in the SEAL framework.

## 6  Conclusion

We propose SEAL, a framework that can adopt a structured energy network as a trainable loss function for training a feedforward network. We show that SEAL is a general framework that can work with various loss functions, such as margin-based, regression-based, and NCE ranking losses, and with different architectures, such as MLPs, CNNs, and pretrained models (BERT). Through extensive experiments on feature-based, text-based MLC, SRL, and image segmentation, we show that SEAL provides an effective way to utilize an energy network and a cross-entropy loss together: performing better, faster, and in a more stable manner when combined. Lastly, through the ablation study, we examine how the loss-net captures the dependencies among output variables. This research opens up doors for various future work, including but not limited to, exploration of different architectures of energy network; application of learned loss-nets in semi-supervised setups. In the long horizon, we are interested in whether general pretrained scoring neural networks, such as BLEURT (Sellam et al., 2020) and BERT-score (Zhang et al., 2019), could serve as a loss function when it can provide backpropagatable gradients to the output space of a feedforward network.

## Acknowledgments and Disclosure of Funding

This work was partially supported by New Faculty Startup Fund from Seoul National University, IBM Research AI through the AI Horizons Network and the Chan Zuckerberg Initiative under the project Scientific Knowledge Base Construction. Additional support was provided by the NSF under Grant Number IIS-2106391, and the Office of Naval Research (ONR) via Contract No. N660011924032 under Subaward No. 123875727 from the University of Southern California.

---

[4]We get label-label dependency from true taxanomy of expr_fun.

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
