# Appendix

## A Gradient to the label representations

As described in section 4, the energy component of loss $L_F$, for the task-net, is the sum of following two parts:

$$E_\Theta^{\text{local}}(\mathbf{x}, \mathbf{y}) = \sum_{i=1}^{L} y_i \mathbf{b}_i^\top T_E(\mathbf{x}) \quad \text{and} \quad E_\Theta^{\text{global}}(\mathbf{x}, \mathbf{y}) = \mathbf{v}^\top \gamma(\mathbf{My}). \tag{11}$$

The gradient of $E^{\text{global}}$ w.r.t $\mathbf{y}$ is given by the following vector of length $L$

$$\frac{\partial E^{\text{global}}}{\partial \mathbf{y}} = \mathbf{M}^T \text{Diag}(\gamma'(M\mathbf{y}))\mathbf{v}. \tag{12}$$

While the gradient of cross entropy $\text{BCE}(y, y^*)$ w.r.t. $\mathbf{y}$ is given by the following vector of length $L$

$$\frac{\partial \text{BCE}}{\partial \mathbf{y}} = \mathbf{y}^* \otimes \frac{1}{\mathbf{y}} - (1 - \mathbf{y}^*) \otimes \frac{1}{1 - \mathbf{y}}, \tag{13}$$

where $\otimes$ is element-wise product.

Note the dependency of the gradient for a particular component $y_i$ of $y$ on other component in the expressions above. The gradient of cross entropy ($\frac{\partial l}{\partial y_i}$) only depends on its label dimension $i$: $y_i$ and $y_i^*$. In contrast, in the case of the energy function as loss (SEAL), the gradient $\frac{\partial L_F}{\partial y_i}$ depends on all $L$ dimensions of $\mathbf{y}$ due to the product term $\mathbf{My}$ in $\frac{\partial E^{\text{global}}}{\partial y_i}$.

## B Task Details

This Section provides task-specific information including task definitions, descriptions for datasets and models, and additional observations in our experiments. Please refer to the beginning of Section 4 for a preliminary description.

### B.1 Multi-label classification

**Task Definition:** Given input $x$, multi-label classification maps the input to the labels $y$ in the output space, $\mathcal{Y} = \{0, 1\}^L$.

**Dataset:** In our experiments, we use 11 multi-label classification (MLC) datasets covering various label spaces, label sizes (ranging from 27 to 4k), training sizes, and input characteristics (binary, real, text) as shown in Table 5. The first seven rows are binary- or real-valued feature-based datasets. In addition to including Bibtex and Delicious that were used in baseline approaches (Belanger & McCallum, 2016; Gygli et al., 2017; Tu & Gimpel, 2018), we carefully picked smaller (Genbase, Cal500) and larger datasets (Eurlex-ev) as well as datasets with known taxonomy (Spo_FUN, Expr_FUN) to examine SEAL in various scenarios. For text-based datasets, we select RCV Lewis et al. (2004) and BGC (Aly et al., 2019) which have label dependencies (DAG and forest respectively), AAPD to compare with state-of-the-art text-based MLC model (Zhang et al., 2021), and take a subset of the NYT dataset Sandhaus (2008) [5] to experiment with larger label dimensions and training data. For text-based MLC datasets, we examine whether SEAL can be helpful in the context of large pre-trained models as well. The links to download the multi-label datasets used in this work are given in Table 6.

**Feature network:** We use standard feed-forward neural network and pretrained BERT-base as the feature network, $T_E$, for the feature-based and text-based MLC datasets respectively.

---

[5] Due to their enormous size, we use a subset of NYT training set, which is still the largest dataset in our experiments.

Table 5: Statistics of the datasets used in the experiments.

| Dataset | Domain | #Instances | | Input Dim | #Labels | Input Type |
|---------|--------|------------|------|-----------|---------|------------|
| | | Train | Test | | | |
| Genbase | Biology | 398 | 132 | 1186 | 27 | Binary |
| Cal500 | Music | 283 | 114 | 68 | 174 | Real |
| Spo FUN | Gene Ontology | 1,600 | 1,266 | 86 | 500 | Real |
| Expr FUN | Gene Ontology | 1,636 | 1,288 | 561 | 500 | Real |
| Bibtex | Text | 4,407 | 1,497 | 1836 | 159 | Binary |
| Delicious | Text | 9,690 | 3,194 | 500 | 983 | Binary |
| Eurlex-ev | Text | 11,557 | 3,881 | 5000 | 3,993 | Real |
| RCV | Text | 13,890 | 781,265 | N/A | 104 | Raw Text |
| AAPD | Text | 53,840 | 1000 | N/A | 54 | Raw Text |
| BGC | Text | 58,715 | 18,394 | N/A | 142 | Raw Text |
| NYT | Text | 175,299 | 180,659 | N/A | 2,109 | Raw Text |

Table 6: Links for downloading the datasets.

| Dataset | Link |
|---------|------|
| AAPD | https://git.uwaterloo.ca/jimmylin/Castor-data/-/tree/master/datasets/AAPD |
| BCG | https://www.inf.uni-hamburg.de/en/inst/ab/lt/resources/data/blurb-genre-collection.html |
| NYT | https://catalog.ldc.upenn.edu/LDC2008T19 |
| RCV1-V2 | http://www.ai.mit.edu/projects/jmlr/papers/volume5/lewis04a/lyrl2004_rcv1v2_README.htm |
| feature based datasets | https://www.uco.es/kdis/mllresources/ |

**Energy network:** Given the input $\mathbf{x}$ and the output $y \in \{0,1\}^L$, the energy terms are defined as:

$$E_\Theta^{\text{local}}(\mathbf{x},\mathbf{y}) = \sum_{i=1}^{L} y_i \mathbf{b}_i^\top T_E(\mathbf{x}), \ E_\Theta^{\text{global}}(\mathbf{x},\mathbf{y}) = \mathbf{v}^\top \gamma(\mathbf{M}\mathbf{y}), \tag{14}$$

where $\mathbf{b}_i, \mathbf{v}, \mathbf{M}, T_E$ contain learnable parameters. $T_E$ is a feature network to represent an input $\mathbf{x}$ in the loss-net, and $\gamma(z) = \log(1 + e^z)$ is the softplus activation function.

### B.2 Semantic Role Labeling

**Task Definition:** Given a sentence and a predicate pair $(W, v)$, the goal of SRL is to identify the spans of arguments for $v$ in $W$ and assign correct role labels to the identified arguments.

**Dataset:** We train and evaluate the performance of SEAL and baselines on the standard benchmark (CoNLL-12) (Pradhan et al., 2013) for SRL.

**Feature network:** We use pretrained BERT-base for the feature network, $T_E$ and $T_F$.

**Energy network:** For the sequence tagging experiments, we take the input, $\mathbf{x}$, a sequence of $N$ tokens and the output, $\mathbf{y}$, a sequence of $N$ labels. $\mathbf{y}_n$ denotes the label at the $n^{th}$ position in the sequence and has $L$ dimensions, the number of labels. Consequently, we define the energy terms as,

$$E_\Theta^{\text{local}}(\mathbf{x},\mathbf{y}) = \sum_{i=1}^{N}\sum_{j=1}^{L} y_{i,j} \mathbf{b}_j^\top T_E(\mathbf{x})_i, \ E_\Theta^{\text{global}}(\mathbf{x},\mathbf{y}) = \sum_{i=1}^{N} \max E_S(\mathbf{y})_i, \tag{15}$$

where $E_S$ applies the self-attention (Vaswani, 2017) over $\mathbf{y}$ to capture the long-distance dependencies among labels and gets a $L$ by $N$ dimension output. We then do max-pooling over the $L$ dimension before taking the sum across the sequence length to get the global energy, $E_\Theta^{\text{global}}$.

### B.3 Image segmentation

**Task Definition:** Given a 3-channel $L$-pixel RGB image $\mathbf{x}$, binary image segmentation requires models to classify each pixel to either foreground or background, thus outputting a binary segmentation mask $\mathbf{y} \in \{0, 1\}^L$.

**Datasets:** The Weizmann Horse Image dataset Borenstein & Ullman (2004) consists of 328 left-oriented horse images annotated with segmentation masks. Following CHOPPS (Li et al., 2013) and DVN (Gygli et al., 2017)[6], we adopt a split of 160 training, 40 validation, and 128 testing examples at the resolution of $32{\times}32$ pixels. We train models using the training split, select hyperparameters according to mean image IoU on the validation split, and report mean image IoU on the test split.

**Feedforward (FCN):** We implement Fully Convolutinal Network described by Gygli et al. (2017) as our feedforward baseline and task-net in SEAL. It has 3 convolutional layers, 2 transposed convolutional layers, and 2 skip connections. The second transposed convolutional layer outputs two channels of logits. A softmax layer is then applied and we take the normalized horse channel as the probability that each pixel belongs to the horse object. The channel can be binarized to obtain the segmentation mask.

**Energy network (DVN):** For segmentation, we do not have separate notions of local and global energies (Eq.(2)). We implement Deep Value Network described by Gygli et al. (2017) as our energy network baseline and loss-net in SEAL, referring to this public repository. In general we find gradient based inference during DVN training and inference unstable and sensitive to configurations like sampling methods and initialization for the segmentation mask. For example, it achieves $< 50$ test IoU if we do not supply ground truth samples during its training. Nevertheless, using the same energy network architecture as loss for FCN under the SEAL-dynamic learning framework improves the performance of FCN. We leave sampling strategies and loss-net architectures for further exploration.

**Qualitative Results:** We observe that segmentation masks predicted by FCN trained with SEAL-dynamic (regression-s) are better at details (e.g. legs) than those by cross-entropy (Figure 3).

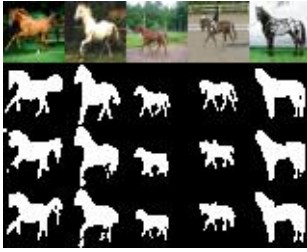

Figure 3: The four rows are the original images, ground truth masks, masks predicted by FCN trained by cross-entropy, and masks by FCN trained with SEAL-dynamic.

## C   Experiment Details

### C.1   Training setup

Our experiments are performed on a mix of TitanX, 1080Ti, 2080ti, M40, and RTX8000 GPUs. We use separate ADAM optimizers (Kingma & Ba, 2014) for the loss-net ($\Theta$) and the task-net ($\Phi$) which optimize parameters in alternating fashion[7]. For each minimization loop, we take $n_E$, $n_F$ gradient steps respectively for estimating $\Theta_t$ and $\Phi_t$. The best hyper-parameters for each model are found using Bayesian search[8]. We also report the training time, inference time, and parameter size of different MLC methods in Appendix D.

---

[6]Script that includes links to and processes the dataset: https://gist.github.com/gyglim/714cb24a1c34c95e0a0c9a8a4ec0620c

[7]The code we used to train and evaluate our models is available a https://anonymous.4open.science/r/SEAL/README.md

[8]We use Weights & Biases (Biewald, 2020) for hyperparameter search

## C.2 Hyperparameters

The following tables tabulate the hyper-parameters for SEAL on each dataset. We fix the $\lambda_2 = 1$ in equation 1 to reduce degree of freedom in hyperparameter search. In MLC and SRL experiments, we fix the task-net number of layers, layer-dimensions, and dropout for the SEAL framework according to the best run of the task-net with cross-entropy. In segmentation experiments, we run sweeps of about 200 runs and use the same search space for overlapping hyperparameters for different models for fair comparisons; once the best hyperparameters are found by wandb sweeps, we use them to run with 5 or 10 random seeds depending on the task. We tested on 10 random seeds for both feature-based MLC and binary imagae segmentation where the models are trained from scratch. For text-based MLC and SRL, we tested on 5 random seeds as running multiple experiments on large pre-trained models such as BERT-base is expensive.

### C.2.1 Multilabel Classification

| method | $n_E$ | $n_F$ | $\lambda_1$ | num samples | $\Theta_{lr}$ | $\Phi_{lr}$ |
|---|---|---|---|---|---|---|
| margin | 1 | 10 | 0.001 | n/a | 0.0002 | 0.001 |
| regression | 6 | 10 | 0.002 | n/a | 0.0003 | 0.001 |
| regression-s | 3 | 10 | 0.03 | 20 | 0.0004 | 0.001 |
| NCE ranking | 9 | 10 | 0.002 | 100 | 0.0005 | 0.001 |

Table 7: genbase

| method | $n_E$ | $n_F$ | $\lambda_1$ | num samples | $\Theta_{lr}$ | $\Phi_{lr}$ |
|---|---|---|---|---|---|---|
| margin | 1 | 10 | 0.001 | n/a | 0.0005 | 0.008 |
| regression | 3 | 10 | 1.4 | n/a | 0.006 | 0.007 |
| regression-s | 9 | 5 | 9 | 5 | 0.008 | 0.006 |
| NCE ranking | 9 | 10 | 0.4 | 40 | 0.007 | 0.005 |

Table 8: cal500

| method | $n_E$ | $n_F$ | $\lambda_1$ | num samples | $\Theta_{lr}$ | $\Phi_{lr}$ |
|---|---|---|---|---|---|---|
| margin | 1 | 1 | 0.0002 | n/a | 0.002 | 0.004 |
| regression | 3 | 5 | 8 | n/a | 0.0001 | 0.00015 |
| regression-s | 6 | 1 | 9 | 5 | 0.0001 | 0.003 |
| NCE ranking | 6 | 5 | 0.9 | 20 | 0.002 | 0.001 |

Table 9: delicious

| method | $n_E$ | $n_F$ | $\lambda_1$ | num samples | $\Theta_{lr}$ | $\Phi_{lr}$ |
|---|---|---|---|---|---|---|
| margin | 1 | 5 | 0.001 | n/a | 0.0003 | 0.002 |
| regression | 6 | 5 | 0.002 | n/a | 0.0006 | 0.004 |
| regression-s | 12 | 1 | 0.01 | 20 | 0.0004 | 0.007 |
| NCE ranking | 6 | 5 | 0.5 | 40 | 0.001 | 0.002 |

Table 10: eurlex

| method | $n_E$ | $n_F$ | $\lambda_1$ | num samples | $\Theta_{lr}$ | $\Phi_{lr}$ |
|---|---|---|---|---|---|---|
| margin | 1 | 5 | 0.01 | n/a | 0.00003 | 0.001 |
| regression | 9 | 5 | 0.02 | n/a | 0.0003 | 0.0006 |
| regression-s | 9 | 5 | 2 | 30 | 0.007 | 0.002 |
| NCE ranking | 6 | 10 | 4 | 80 | 0.00005 | 0.0002 |

Table 11: expr_fun

| method | $n_E$ | $n_F$ | $\lambda_1$ | num samples | $\Theta_{lr}$ | $\Phi_{lr}$ |
|---|---|---|---|---|---|---|
| margin | 1 | 10 | 0.08 | n/a | 0.0001 | 0.002 |
| regression | 1 | 5 | 9 | n/a | 0.001 | 0.00015 |
| regression-s | 9 | 10 | 0.02 | 10 | 0.0002 | 0.0009 |
| NCE ranking | 3 | 10 | 0.7 | 60 | 0.008 | 0.0005 |

Table 12: spo_fun

| method | $n_E$ | $n_F$ | $\lambda_1$ | num samples | $\Theta_{lr}$ | $\Phi_{lr}$ |
|---|---|---|---|---|---|---|
| margin | 1 | 6 | 0.0002 | n/a | 0.00005 | 0.001 |
| regression | 9 | 1 | 3 | n/a | 0.005 | 0.003 |
| regression-s | 5 | 5 | 9 | 10 | 0.005 | 0.001 |
| NCE ranking | 12 | 5 | 5 | 20 | 0.0004 | 0.001 |

Table 13: bibtex

| method | $n_E$ | $n_F$ | $\lambda_1$ | num samples | $\Theta_{lr}$ | $\Phi_{lr}$ |
|---|---|---|---|---|---|---|
| NCE ranking | 1 | 1 | 1 | 300 | 0.00001 | 0.0001 |

Table 14: AAPD

### C.2.2 Semantic Role Labeling

We set batch size to 16 and weight decay to 0.00001 for both loss-net ($\Theta$) and the task-net ($\Phi$). For the self-attention used in loss-net ($\Theta$), we set the hidden dimension to 256 and number of attention head to 1. We again fix $n_E$, $n_F$ as 1.

| method | $n_E$ | $n_F$ | $\lambda_1$ | num samples | $\Theta_{lr}$ | $\Phi_{lr}$ |
|---|---|---|---|---|---|---|
| NCE | 1 | 1 | 0.5 | 400 | 0.00001 | 0.00001 |

Table 15: CoNLL-12

### C.2.3 Weizmann Horse Segmentation

In all experiments, we use batch size 8 and 36 image crops (Gygli et al., 2017). In DVN with GBI, the found learning rate for GBI to get adversarial samples is 7.82 while that for other GBI is 6.96.

| method | $n_E$ | $n_F$ | $\lambda_1$ | num samples | $\Theta_{lr}$ | $\Theta_{wd}$ | $\Phi_{lr}$ | $\Phi_{wd}$ |
|---|---|---|---|---|---|---|---|---|
| FCN (cross-entropy) | n/a | n/a | n/a | n/a | n/a | n/a | 2.01e-3 | 5.86e-5 |
| DVN with GBI | n/a | n/a | n/a | n/a | 2.73e-4 | 4.73e-4 | n/a | n/a |
| SEAL-dynamic (regression) | 8 | 10 | 0.20 | n/a | 1.9e-2 | 1.41e-4 | 3.64e-4 | 3.75e-5 |
| SEAL-dynamic (NCE) | 1 | 1 | 0.0002 | 500 | 1.01e-5 | 4.17e-4 | 3.96e-3 | 3.59e-5 |

Table 16: Weizmann Horse segmentation. lr is learning rate. wd is weight decay.

# D  Analysis: Parameter Size and Speed of Inference and Training

We present the analysis on number of parameters (Table 17), inference speed (Table 18), and training speed (Table 19) that different methods utilize for Multilabel Classification datasets. Theses experiments were performed on TitanX GPU with 14GB CPU memory.

In summary, SEAL requires approximately twice number of parameters compared to cross-entropy loss on training time, however it requires the same amount of parameter on inference time. Similarly, train time (sec) per epoch of SEAL is slower than that of cross-entropy and energy network as SEAL requires multiple backpropagation steps for both energy network and feedforward network. Nonetheless, again, at inference time, the speed of CE and SEAL is the same and is much faster (2×-7×) than the inference of energy network.

The analysis we present below are on the seven feature-based dataset, however the trend is similar for text-based dataset as well: twice parameter size on training time but equal parameter and runtime in inference time with CE method. The only difference in the text-based MLC is, due to running on a much larger BERT-baed models, that we only run one step for each update of feedforward network and energy network. Thus, unlike Table 19, train time of SEAL is approximately twice that of CE.

| Dataset \ Methods | Parameter size | | | |
| --- | --- | --- | --- | --- |
| | CE | Energy network (SPEN, DVN, NCE) | SEAL (Train-time) | SEAL (Inference-time) |
| EXPR_FUN | 333000 | 533800 | 866800 | 333000 |
| SPO_FUN | 447000 | 597600 | 1044600 | 447000 |
| Bibtex | 958800 | 991000 | 1949800 | 958800 |
| Cal500 | 1158700 | 1123500 | 2282200 | 1158700 |
| Delicious | 754000 | 951000 | 1705000 | 754000 |
| Genbase | 485600 | 491400 | 977000 | 485600 |
| Eurlex-ev | 4747500 | 5546500 | 10294000 | 4747500 |

Table 17: The number of parameters required during train time for SEAL is approximately double the size of feedforward (CE column) while energy network and feedforward sizes are comparable. However, in the inference time, SEAL has an equal amount of parameters to the CE column as only feedforward network is utilized during inference time.

| Dataset | Inference time (sec) | | Inference speed (examples/sec) | | Speed ratio (CE and SEAL/ Energy Network) |
| --- | --- | --- | --- | --- | --- |
| | Energy network | CE and SEAL | Energy network | CE and SEAL | |
| EXPR_FUN | 1.33 | 0.22 | 638 | 3801 | 5.96 |
| SPO_FUN | 1.18 | 0.16 | 709 | 5231 | 7.38 |
| Bibtex | 3.6 | 1.78 | 414 | 840 | 2.03 |
| Cal500 | 0.24 | 0.10 | 438 | 1080 | 2.47 |
| Delicious | 5.35 | 1.39 | 599 | 2307 | 3.85 |
| Genbase | 0.23 | 0.13 | 574 | 1005 | 1.75 |
| Eurlex-ev | 24 | 12.22 | 162 | 317 | 1.96 |

Table 18: We simply average inference time for CE and SEAL variants as they are very similar. Likewise, we average the inference time of different energy networks. Here, inference time (sec) is recorded for the whole validation set. We also present speed per sec (example/sec) and speed ratio. The last column shows that CE and SEAL methods are 2x-7x faster in inference time than the energy networks.

| Methods\ Datasets | Train time (sec/epoch) | | | | | | |
|---|---|---|---|---|---|---|---|
| | EXPRFUN | SPO_FUN | Bibtex | Cal500 | Delicious | Genbase | Eurlex-ev |
| CE | 6.29 | 5.46 | 22.67 | 1.00 | 33.01 | 2.12 | 114.84 |
| SPEN | 10.99 | 11.01 | 28.04 | 2.53 | 37.31 | 3.57 | 136.10 |
| DVN | 10.95 | 10.02 | 32.12 | 1.87 | 55.24 | 2.79 | 129.94 |
| NCE | 3.71 | 3.83 | 13.97 | 2.82 | 22.03 | 3.68 | 89.33 |
| SEAL-margin | 27.96 | 35.24 | 212.94 | 5.78 | 44.60 | 8.73 | 260.10 |
| SEAL-regression | 46.32 | 56.97 | 27.78 | 8.07 | 143.90 | 13.91 | 352.63 |
| SEAL-regression-s | 42.87 | 77.92 | 179.97 | 8.48 | 45.33 | 10.79 | 221.33 |
| SEAL-NCE | 72.43 | 40.34 | 131.73 | 16.58 | 158.97 | 17.77 | 431.92 |
| SEAL-Ranking | 41.24 | 26.83 | 218.37 | 7.04 | 317.63 | 10.74 | 408.79 |

Table 19: The training time per epoch is presented per dataset and per loss function used. Due to different gpu types and node status, there are some outliers. Furthermore, as SEAL runs multiple number of backpropagation steps for energy network and feedforward network in the alternating optimization, direct comparison of training time is not really available. However, the general trend in terms of training time per epoch is CE < Energy Networks < SEAL.

# E Further analysis continued

## E.1 Effect of samples

Given the notable performance of the NCE ranking loss, we raise the question of whether drawing many samples from the task-net helps better estimation of the energy surface. To examine this question, we perform additional experiments using the regression loss with sampling. Given a sample set $S$, we define the regression-s loss as $L_{E-\text{regression-s}} = \sum_{\tilde{\mathbf{y}} \in S} L_{E-\text{regression}}(\mathbf{x}, \mathbf{y}^*, \tilde{\mathbf{y}}; \Theta)$. We take two approaches in collecting these samples: (1) discrete binary vectors drawn from probability vector $\tilde{\mathbf{y}}$ as done with the NCE ranking loss and (2) continuous perturbation of $\tilde{\mathbf{y}}$ with Gaussian noise. We find that discrete samples do not make any notable changes between regression and regression-s, while continuous samples result in +0.4 F1 improvements on average. We report results of regression-s using continuous samples in row 10 of Table 1. We conclude that sampling helps capturing better energy surfaces for training the task-net. However, the effect and characteristics of the samples might differ for different types of energy losses. We believe sampling is more effective for the NCE ranking loss where the group of samples contribute in a relative manner in learning surface, in contrast to the regression-s loss to which each sample contributes independently.

## E.2 Data-specific analysis

We analyze the characteristics of selected datasets to reason about the high performances of SEAL on them.

### E.2.1 genbase

Here, we analyze the factors behind SEAL achieving almost perfect F1 score on the genbase dataset. It seems genbase stands out with its small label size (27 which is the smallest among Table 5) and with very clear pattern in the label space. Upon analysis, out of 27 labels, we found that 6 labels only occur as a singleton (by itself), 10 labels only occur as non-singleton, and 7 labels do not participate at all. This leaves only 4 labels occurring by themselves as well as with others. Not only that, only 7% (35/500) of training instances have more than one active label. In this peculiar setting, we conjecture feedforward networks with cross-entropy (CE) loss can easily learn singletons, and energy networks would easily learn co-occurrence, but learning both might be confusing for these two models at the opposite end of the spectrum. We believe the synergy of CE and a loss from the energy network enables capturing the best of both worlds and achieving a near-perfect score that none of the approaches by itself achieves. As can seen in Table 1, neither cross-entropy nor energy networks are nearly as strong as SEAL methods.

### E.2.2 Delicious and Cal500

In Table 1, it seems SEAL-NCE and SEAL-Ranking are particularly well performing on cal500 and delicious. It turns out that cal500 and delicious have a very high diversity of 1 and 0.981 (Reference:

). Diversity of 1 means that each data point holds a unique label set. It seems that the exposure to multiple samples and evaluations of their relative scores in the NCE ranking loss function can be helpful in a high-diversity setting.

# F  Additional Experiment: Multi-label Classification on Text Dataset with BERT Adapters

Table 20: Test F1 for text MLC datasets.

| method \ datasets | BGC | RCV | NYT |
|---|---|---|---|
| cross-entropy | 81.15 | 87.18 | 77.4 |
| SEAL-dynamic with margin (InfNet) | 81.14 | 87.01 | 78.13 |
| SEAL-dynamic with NCE | **81.64** | **87.82** | **78.87** |

To test the effectiveness of SEAL on training large language models, in Table 20, we further experiment with text datasets using pre-trained BERT (Devlin et al., 2019) with adapter (Houlsby et al., 2019; Pfeiffer et al., 2020) as the feature network $T_E$ and $T_F$. Considering the computational expense of running hyper-parameter searches on BERT, based on Table 1, we choose the best-performing SEAL (SEAL-dynamic with NCE ranking) to compare with baselines: InfNet and cross-entropy. Table 20 shows that SEAL-dynamic with the NCE ranking loss is effective on pre-trained models that utilizes large text datasets as well. On average, SEAL-NCE gains 0.87 and 0.68 F1 points over cross entropy and SEAL-margin (InfNet) respectively on the text datasets.

# G  Gradient Plots

In multi-label classification, the global energy $E_\Theta^{\text{global}}(\mathbf{y})$, one of the two constituents of the loss-net, captures the dependence between different labels. In order to inspect the nature of the dependence captured, we randomly pick a tuple of labels $(l_i, l_k)$ such that $l_i \bowtie l_k$, where $\bowtie$ is a particular relation and plot samples of $\frac{\partial E^{\text{global}}}{\partial y_k}(\mathbf{y})$ in the following manner. First, we sample each component of $\mathbf{y}$ uniformly and independently, which, with a slight abuse of notation can be written as $\mathbf{y} \sim$ Uniform$(0, 1)^L$; then heatmap of $\frac{\partial E_\Theta^{\text{global}}(\mathbf{y})}{\partial y_k}$ on coordinates $(y_i, y_k)$ are plotted. Figure 2b (see Fig. 4, 6 for more examples) shows the samples from $\frac{\partial E^{\text{global}}}{\partial y_k}(y_i, y_k)$ when $i$ and $k$ are such that $y_i \implies y_k$, i.e. there is a positive association between labels $l_i$ and $l_k$. As seen, larger values of $y_i$ makes the gradient w.r.t $y_k$ always higher for the same value of $y_k$ showing positive association between $y_i$ and $y_k$. Contrast this with Figure 2c (see Fig. 5, 7 for more examples), where $i$ and $k$ are such that there is no relationship between $y_i$ and $y_k$. In this setting, $\frac{\partial E_\Theta^{\text{global}}}{\partial y_k}$ does not change w.r.t $y_i$. Furthermore, as expected, Figure 2a, shows that when the same procedure is applied to the cross-entropy loss function, we find no dependence between any pair of labels. For the plots discussed in this section, we analyze energy model trained on DVN (4, 5) and SEAL-dynamic-NCE (6, 7) trained on the expr_fun dataset.

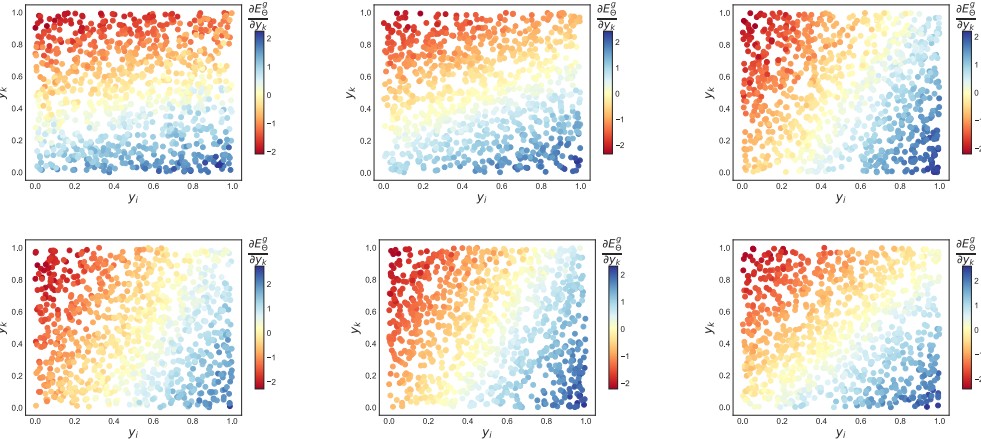

Figure 4: $(y_i, \frac{\partial E_\Theta^{global}(\mathbf{y})}{\partial y_k})$ when $y_i \implies y_k$. Analysis of energy model trained with DVN.

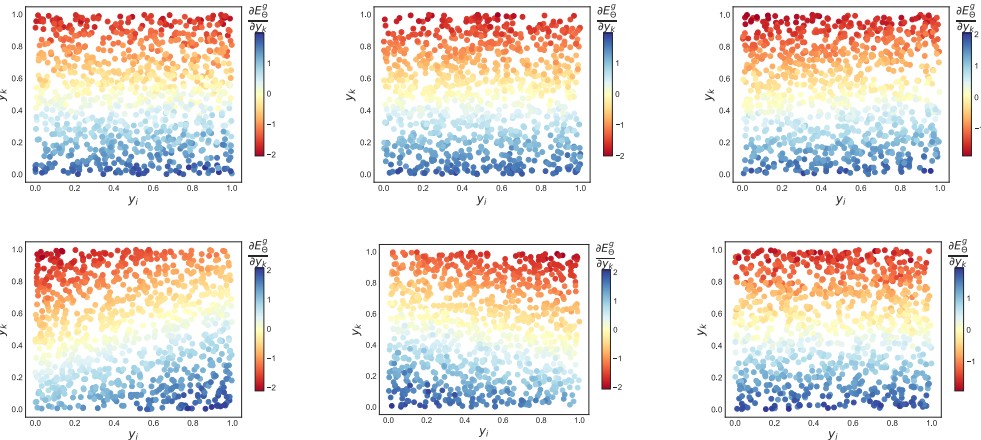

Figure 5: $(y_i, \frac{\partial E_\Theta^{global}(\mathbf{y})}{\partial y_k})$ when there is no relation between labels $l_i$ and $l_k$. Analysis of energy model trained with DVN.

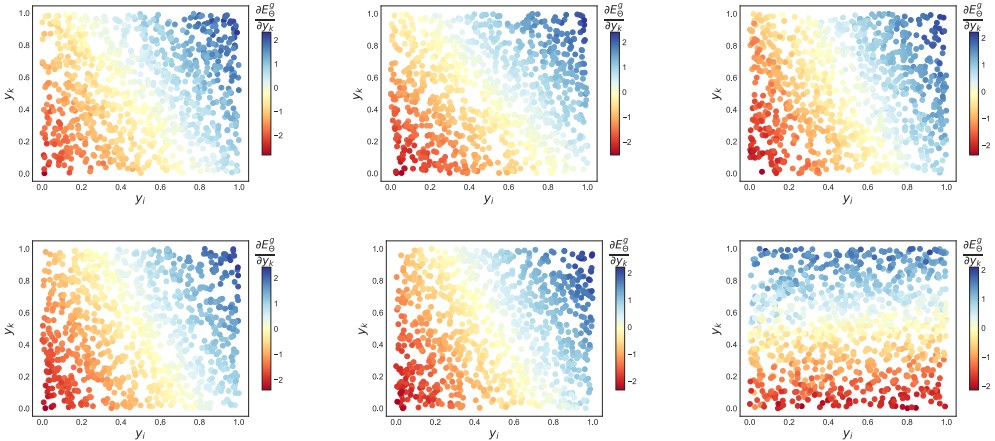

Figure 6: $(y_i, \frac{\partial E_\Theta^{global}(\mathbf{y})}{\partial y_k})$ when $y_i \implies y_k$. Analysis of energy model trained with SEAL-dynamic NCE.

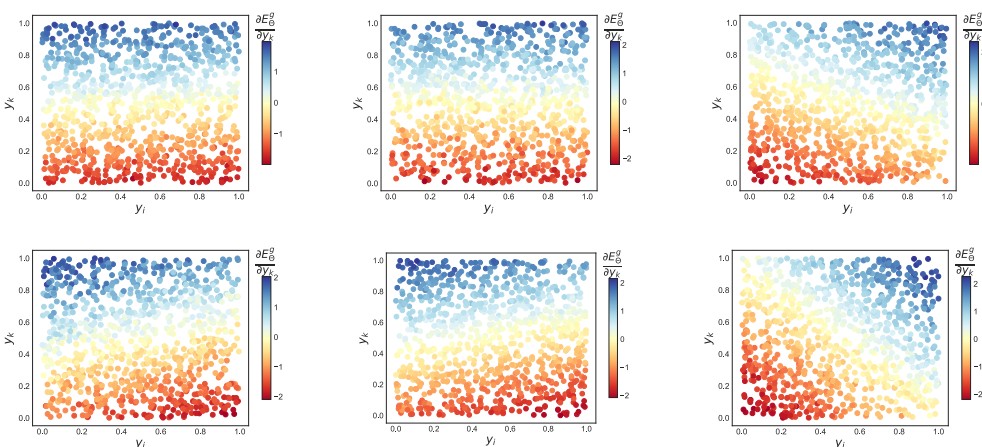

Figure 7: $(y_i, \frac{\partial E_\Theta^{global}(\mathbf{y})}{\partial y_k})$ when there is no relation between labels $l_i$ and $l_k$. Analysis of energy model trained with SEAL-dynamic NCE.