# OpenReview forum: "Structured Energy Network As a Loss"
_NeurIPS.cc/2022/Conference — NeurIPS 2022 Accept_

### Official Review · Reviewer_Y7gJ · 2022-07-07

**Rating:** 7
**Confidence:** 4
**Soundness:** 3 good
**Presentation:** 3 good
**Contribution:** 3 good

**Summary:**

The paper proposes the combined use of structured prediction energy networks (loss-net) as a loss function that considers relations in the output. This loss function is then used to train a feed forward network (task-net) to speed up inference at test time. They proposed both a static and dynamic approach to train these two networks.

**Questions:**


- The loss function of task-net is a weighted average with parameters $\lambda_1$ and $\lambda_2$, the authors fix the latter to 1 and perform bayesian search to determine the best $\lambda_1$. The variability of this parameter is high and for certain tasks or energy losses it sets to be quite small. For example, Table 7 in Appendix C.2.1 shows that for genbase (the best performing task for the presented model) the influence of loss-net is quite small compared to simple BCE. Nonetheless, the model performs substantially better than CE. Could authors provide an intuition of why this is happening? It seems to be consistently true for margin as an energy loss across tasks and there are other interesting patterns for different tasks. A discussion on this might help in future usability of the proposed approach.
- Could the author provide confidence intervals for the results?
- Given the closeness of the results for SEAL-static and SEAL-dynamic which one would the authors recommend for use? Is there any advantage in one above the other based on context?
- What’s the behavior in training of SEAL-dynamic? I imagine that given the alternating minimization approach training would be highly variable in early stages. Does it require more epochs to be trained due to its alternating minimization nature? Could the author provide details in this regard? I could not find any mention of it either in the main text nor the appendix.
- Similar to the question above, how is batch size selected? It seems it should be a crucial parameter in stabilizing training and performances but it is not mentioned.


**Limitations:**

See questions.

**Strengths And Weaknesses:**

The paper is well written and easy to follow, it also does a good job in placing itself in the literature and, as the authors state themselves, it is a generalization of previous work [Tu et al, 2020] aimed at reducing inference time for energy networks. Results show remarkable improvement over the state of the art.

---

> ### Author Response · Authors · 2022-07-31
> **Thank you for your time and thoughtful review.**
>
> Thank you for your time and thoughtful review.
>
> > The loss function of task-net is a weighted average with parameters λ1 and λ2, the authors fix the latter to 1 and perform bayesian search to determine the best λ1. The variability of this parameter is high and for certain tasks or energy losses it sets to be quite small.
>
> The loss functions from cross entropy and energy network can come from wildly different function families and hence can have very different shape and steepness. For this reason, the weights for these losses cannot be compared on the same scale. Moreover, because the loss function coming from energy is learnt, there can be high variability in $\lambda_1$ based on data and task. We agree that gaining further understanding of the loss functions coming from the energy network is important and leave it as future work.
>
> > For example, Table 7 in Appendix C.2.1 shows that for genbase (the best performing task for the presented model) the influence of loss-net is quite small compared to simple BCE. Nonetheless, the model performs substantially better than CE.
> >
>
> As we stated above, it’s hard to exactly decompose the effect of loss-net. That being said, noticing the out-of-norm behavior of genbase, we analyze its peculiar data distribution in appendix E.3.1.
>
> > Given the closeness of the results for SEAL-static and SEAL-dynamic which one would the authors recommend for use? Is there any advantage in one above the other based on context?
> >
>
> We recommend SEAL-dynamic as it usually performs better and as it also simplifies the engineering procedure because SEAL-static requires separate training of loss-net prior to training of task-net whereas SEAL-dynamic does not require such procedure and can start from scratch.
>
> > What’s the behavior in training of SEAL-dynamic? I imagine that given the alternating minimization approach training would be highly variable in early stages. Does it require more epochs to be trained due to its alternating minimization nature? Could the author provide details in this regard? I could not find any mention of it either in the main text nor the appendix.
> >
>
> Owing to the cross entropy term in our loss function in $L_F$ (eq.1), the learning procedure is pretty stable. We did not observe any significant difference between the number of epochs it takes to train SEAL-dynamic vs. cross-entropy loss. While the number of epochs varies by dataset, the average number of epochs is slightly smaller for SEAL-dynamic.
>
> For more specific training-time analysis, please refer to runtimes (Table 19) & model parameter sizes (Table 17) in the appendix.
>
> > how is batch size selected? It seems it should be a crucial parameter in stabilizing training and performances but it is not mentioned.
> >
>
> SEAL-dynamic trains without stability issues on most hyper-parameter settings, including various batch sizes, due to the presence of cross-entropy in the loss. Below we mention the procedure we use to find the hyper-parameters (including the batch size).
>
> 1. We optimized hyper-parameters like feedforward network sizes and batch size on CE models (baseline feedforward models) first and fixed those hyper-parameters for other models giving an advantage to the baseline.
> 2. After that, we only searched for additional hyper-parameters like loss-net size and loss weight.
>
> Note that even when we do not optimize every possible hyper-parameter for the SEAL framework, this framework outperformed both feedforward and energy network models in a stable manner.

---

### Official Review · Reviewer_iYhN · 2022-07-11

**Rating:** 6
**Confidence:** 4
**Soundness:** 3 good
**Presentation:** 3 good
**Contribution:** 3 good

**Summary:**

Energy based networks have to run gradient based inference (GBI), which can be quite expensive. In this paper, the authors propose to use an energy based network to develop a structured prediction loss. Thereafter (or simultaneously for the dynamic approach), a feedforward network is learned that tries to match the energy network's output (modeling structure) and a cross-entropy loss w.r.t. ground truth outputs. Combined, the proposed technique gives the inference efficiency of feedforward networks, yet is able to model structure similar to an energy network.


**Questions:**

* Is the InfNet model in Table 1 from Tu et al. (2020)? That is, is the updated and closer to the current method version of the baseline being used for comparison?

* How expensive is SEAL's training? How does it fare in comparison to other methods? I suspect that twice the model size (due to a Feedforward network and an energy based network) plus having to run GBI to get the energy-based network's output will yield a significant training cost.

* Alternating minimization optimization problems can often be sensitive and hard to tune. Did that cause any difficulty for the dynamic model? Regardless, I think this point should be discussed.


#### Minor comments:

* The global energy in Equation 2 does not have a dependence on x. Is this correct?

**Limitations:**

It will be valuable to discuss the limitations of using a feedforward network to "learn to do GBI".

**Strengths And Weaknesses:**

### Strengths

The proposed method is quite simple, yet effective. The idea is fairly obvious, but getting the details right is laudable. The paper is decently well-written and well-motivated.

### Weaknesses

* Limitations should be better discussed: The use of a feedforward network to map the input to the GBI output of the energy based network can be thought of as "learning to do gradient-based inference". It is unclear to what extent the strategy can be expected to generalize.

* Relation with InfNet: The authors argue InfNet is a special case of their proposed method. While I understand that there is a train-test mismatch for InfNet, the follow-up by Tu et al. (2020) where two separate networks are used for learning an adversarial sampler and for performing inference gets very close to the proposed method. It would be helpful to focus more on the discussion separating these methods. Currently, the discussion makes it seem like the proposed SEAL method is akin to a slight extension.

Post rebuttal: The authors justifiably addressed my concerns and I have accordingly raised my scores.

---

> ### Author Response · Authors · 2022-07-31
> **Thank your for taking your time and providing thoughtful feedback.**
>
> Thank your for taking your time and providing thoughtful feedback.
>
> Before answering specific questions, we wanted to clarify the central premise of our paper: “energy network as a loss”.  Please note that SEAL-dynamic does not require any kind of gradient-based inference (GBI) steps, neither in training nor in inference time. The energy network (loss-net) provides a differentiable loss function that can be use to backprop all the way to the feedforward network.
>
> To put the previous work in context, Tu et. al. claimed that the feedforward network is mimicking inference-time behavior of energy network, which leads one to believe that one first obtains the GBI output $\mathbf{\hat y}$ and then makes the feedfoward output $\mathbf{\tilde y}$ follow $\mathbf{\hat y}$. However, this is not what happens in Tu et al., nor in our paper as the energy network is used as loss to update feedforward network without calculating $\mathbf{\hat y}$. By reinterpreting energy network as  loss, we enable SEAL to utilize various different loss formulations and be not limited by adversarial formulation of InfNet.
> > Limitations should be better discussed: The use of a feedforward network to map the input to the GBI output of the energy based network can be thought of as "learning to do gradient-based inference". It is unclear to what extent the strategy can be expected to generalize.
> >
>
> As we stated above, we want to clarify that we are not learning to perform GBI, but instead we are utilizing energy network as a learnable loss function that can capture dependency across structured multivariate output.
>
> We believe SEAL framework can be applied to any structured prediction problem where we can relax the discrete output space to a probability simplex. We also showed that our method can be applied to different network structures (MLP, BERT, CNN), to different tasks, and to different data characteristics. If these are not the axis of generalization that the reviewer was looking for, please let us know so we can better address your concerns.
>
> > Is the InfNet model in Table 1 from Tu et al. (2020)? That is, is the updated and closer to the current method version of the baseline being used for comparison?
>
> Yes, it is 2020 model with cost-augmented layer.
>
> > Relation with InfNet
>
> What we tried to establish in our paper in relation to InfNet is two folds.
>
> 1. We are generalizing as we show that loss for feedforward network, -E(x,y), and energy loss does not have to hold specific relationship such as an adversarial relationship used in Tu et al., 2019, 2020, i.e.  $\min_\Theta \max_\Phi f(x,y;\Theta, \Phi)$ .  This opens up many other loss functions to be applied into our propoesd SEAL framework as we show with the regression and NCE loss.
> 2. We are reinterpreting the mechanism of InfNet. As we stated above, we wanted to clearly identify what is the success behind InfNet. We hypothesized the reason of success was because the energy network was used as a loss function and provided experimental results  that supports our hypothesis. Experiments with SEAL-static especially well exhibits this behavior as loss-net is trained once and fixed as a  loss function.
>
> > How expensive is SEAL's training? How does it fare in comparison to other methods? I suspect that twice the model size (due to a Feedforward network and an energy based network)
>
> We compare the SEAL-dynamic’s runtimes (Table 19) take & model parameter sizes (Table 17) in the appendix. It is correct that it takes longer on train time and more memory. However, at inference time it is faster than energy network that runs GBI and takes the same memory and same speed as the conventional feedforward network.
>
> > plus having to run GBI to get the energy-based network's output will yield a significant training cost.
>
> We do not run GBI during training of feedforward network in SEAL framework. Please refer to the first two paragraphs of this response for details.
>
> For SEAL-static, we do have extra memory consumption for loss-net. Please refer to the Table 17, 19.
>
> > Alternating minimization optimization problems can often be sensitive and hard to tune. Did that cause any difficulty for the dynamic model? Regardless, I think this point should be discussed.
>
> We believe the cross-entropy term in the loss term stabilizes the learning process. Because of this we did not observe any stability issues in SEAL dynamic.
>
> > The global energy in Equation 2 does not have a dependence on x. Is this correct?
>
>
> That is correct. This is the same equation used in Belanger and McCallum (2016), Gygli et al. (2017), and Tu et al., (2019) for multi-label classifcation. The local term takes care of energy between input x and each label $y_i$, while the global energy term takes care of energy across multivariate $y_1, y_2, \dots, y_L$.
>
> The only exception to this formulation was binary image segmentation following Gygli et al. (2017) for a fair comparison.

---

> > ### Comment · Reviewer_iYhN · 2022-08-02
> > **Response to rebuttals**
> >
> > Thank you for the detailed responses. My questions were primarily based on the added complexity of GBI for model training and the provided clarifications are really useful. I see the significance of the proposed technique and have accordingly raised my score.
> >
> > However, I want to stress that it will be very useful to add the clarification responses in your rebuttals to the main paper. For example, adding discussions on why alternating minimization seems to be stable because of the cross-entropy loss (it would be useful to see sensitivity study w.r.t. $\lambda_1$ for that), a forward reference to the training time tables or direct inclusion in the main paper, better discussion on relation with InfNet, etc. For me, the comments were useful to helping understand the paper's impact and may help other readers as well.

---

> > > ### Author Response · Authors · 2022-08-04
> > > **Thank you for your prompt response.**
> > >
> > > Thank you for your prompt response! We will reflect your suggestions into our camera ready.

---

### Official Review · Reviewer_cyaU · 2022-07-11

**Rating:** 7
**Confidence:** 3
**Soundness:** 3 good
**Presentation:** 3 good
**Contribution:** 3 good

**Summary:**

The authors propose using the structured energy network as a loss to achieve faster and better performance on structured prediction problems. They demonstrate that their approach achieves competitive results, where a dynamic component of the loss can adapt to specific characteristics of the problem at hand.

**Questions:**

1) The authors measure the performance of their model using F1 scores for classification. While the results are promising and valid, would it be possible to compute the AUC ROC too? This would provide a measurement agnostic to the cut thresholds, providing a better assessment of the classification performance. Furthermore, clarifications should be provided regarding how the cut threshold is computed to obtain the F1 score.

2) The authors report the mean IoU metric for image segmentation. It would make sense to compute some additional metrics (e.g., https://cocodataset.org/#detection-eval) to make it easier to compare the results against other well-known models? Furthermore, would it make sense to execute additional experiments regarding image segmentation on a well-known benchmark dataset (e.g., PASCAL VOC)?

3) In the description provided for Table 1, please consider: "are marked with underline" -> "are underlined"

4) The authors may be interested in the following research work, at least for the literature review section: Bechtle, Sarah, et al. "Meta learning via learned loss." 2020 25th International Conference on Pattern Recognition (ICPR). IEEE, 2021.

5) In Table 1, the authors report the inference speed, but no information is provided regarding the hardware on which such models were executed for inference. Could such data be provided? We consider hardware specifications to provide context to put execution times in perspective.

**Limitations:**

We did not find potential negative societal impacts in the present work. The authors did not provide details regarding the limitations of the proposed approach.

**Strengths And Weaknesses:**

The authors present a novel framework where a trainable loss function is used to train a network to perform the inference through a forward pass. The paper is well structured, and the main concepts are clearly conveyed to the reader. The authors created many experiments and considered the ceteris paribus principle to ensure the results could be attributed to a particular change or aspect of their framework, ensuring the quality of research. They demonstrated their approach on various tasks and datasets, obtaining good results. Nevertheless, we miss code to ensure reproducibility or a statement ensuring such code would be made available upon acceptance.

---

> ### Author Response · Authors · 2022-07-31
> **Thank your for taking your time and providing thoughtful review.**
>
> Thank your for taking your time and providing thoughtful reviews.
>
> > Nevertheless, we miss code to ensure reproducibility or a statement ensuring such code would be made available upon acceptance.
>
> A [link]([https://anonymous.4open.science/r/SEAL/README.md](https://www.google.com/url?q=https://anonymous.4open.science/r/SEAL/README.md&sa=D&source=docs&ust=1658967677872626&usg=AOvVaw3EnzLD_QGBB6UeaCRw_-jp)) to the anonymized code is provided in our submitted draft (footnote 3). We will also make our code public upon acceptance.
>
>
> > IOU metric vs. other metrics
> >
>
> COCO metrics evaluate the precision and recall of multiple predicted objects per image in the task of instance detection and segmentation, where each image usually contain several objects; they are not suitable for the Weizmann Horse task, which is a binary segmentation task of a single object per image. For fine-grained evaluation of segmentation quality in the Weizmann Horse task, we follow DVN to calculate mean IoU between ground truth and predicted masks, unlike COCO evaluation during which IoU values are only utilized to filter out (segmentations with IoU values under a certain threshold) frivolous predictions.
>
> > Furthermore, would it make sense to execute additional experiments regarding image segmentation on a well-known benchmark dataset (e.g., PASCAL VOC)?
> >
>
> We defer the study of task-net and loss-net architectures for more datasets including PASCAL VOC to future work, due to limited resources and as we think the set of experiments we present is already a handful to convey the effectiveness of SEAL framework.
>
> > In Table 1, the authors report the inference speed, but no information is provided regarding the hardware on which such models were executed for inference. Could such data be provided? We consider hardware specifications to provide context to put execution times in perspective.
> >
>
> We used TitanX with 14GB CPU memory for the inference speed reports and for feature-based MLC dataset. For other experiments, we use GPUs with larger memory like M40 and RTX8000.

---

> ### Author Response · Authors · 2022-07-31
> **Regarding F1 calculation.**
>
> > The authors measure the performance of their model using F1 scores for classification. While the results are promising and valid, would it be possible to compute the AUC ROC too? This would provide a measurement agnostic to the cut thresholds, providing a better assessment of the classification performance.Furthermore, clarifications should be provided regarding how the cut threshold is computed to obtain the F1 score.
>
>
> For F1 calculation, we use a fixed threshold of 0.5 for all the models, following previous literature of SPEN, DVN, and InfNet.
>
> We compute the ‘samples’ version of [average precision score](https://scikit-learn.org/stable/modules/generated/sklearn.metrics.average_precision_score.html
> ), which is a threshold-free metric similar to AUC ROC.  As noted [here](https://scikit-learn.org/stable/modules/generated/sklearn.metrics.average_precision_score.html
> ), for the ‘samples’ version, the mean of average precision values is computed across samples. The table below, which we plan to add to the appendix, shows the mean average precision for various models. Best results per dataset are annotated with asterisks; best results within a framework (energy network with GBI, SEAL-static, SEAL-dynamic) are marked in bold. Note that the models were not chosen with the best MAP score but chosen with the best F1 validation score.
>
> ### MAP Performance for feature-based MLC datasets
>
> |                   |  Use of samples | **bibtex** | **delicious** | **genbase** | **cal500** | **eurlexev** | **expr_fun** | **spo_fun** | **Average** |
> |-------------------|---|:----------:|:-------------:|:-----------:|:----------:|:------------:|:------------:|:-----------:|-------------|
> | **cross-entropy** | x |     54.95 |        37.24 |      75.61 |     50.59 |       47.39 |      * **47.42** |      40.13 |    50.47   |
> |    energy only    |   |            |               |             |            |              |              |             |             |
> | **SPEN**          | x |     35.07 |    **25.36** |      42.75 | **36.93** |   **38.25** |   **40.05** |  **30.83** | **35.61**  |
> | **DVN**           | x | **36.68** |        17.57 |  **72.13** |     31.53 |       20.02 |       17.85 |      14.03 | 29.97      |
> | **NCE**           | o |      6.81 |         4.99 |      10.98 |     27.22 |        0.13 |       15.16 |       7.03 | 10.33      |
> |    SEAL-Static   |   |            |               |             |            |              |              |             |             |
> | **margin**        | x | **56.15** |    **39.77** |      66.21 |     50.96 |       47.45 |       47.07 |  **39.79** | 49.63      |
> | **regression**    | x |     54.40 |        34.31 |      98.80 |     50.58 |   * **47.65** |   **47.24** |      38.84 |  53.12 |
> | **NCEranking**    | o |     54.55 |        36.36 |  **98.94** | **51.49** |       47.53 |       46.63 |      39.29 |  **53.54** |
> |        SEAL-Dynamic       |   |            |               |             |            |              |              |             |             |
> | **margin**        | x |     55.06 |        36.63 |      98.82 |     49.07 |       40.17 |       46.42 |      37.60 | 51.97      |
> | **regression**    | x |     56.62 |        38.84 |      98.98 |     51.15 |       45.44 |   **47.33** | * **40.17** |  54.08 |
> | **regression-s**  | o | * **56.67** |    * **40.25** |      98.90 | * **51.51** |   **47.16** |       46.56 |      37.76 |  * **54.11** |
> | **NCEraking**     | o |     56.65 |        37.76 |      98.91 |     47.33 |       44.84 |       46.32 |      37.76 | 52.80      |
> | **ranking**       | o |     54.37 |        39.36 |  * **99.05** |     43.36 |       45.75 |       47.16 |      39.29 | 52.62      |
>
>
> ### AAPD MAP Performance
>
> | method | MAP |
> | --- | --- |
> | BERT (cross-entropy) | 82.59 |
> | SEAL-dynamic-NCE | **83** |
>
> ### MAP Performance for other text-based datasets in Appendix
>
> | method \ datasets | BGC | NYT |
> | --- | --- | --- |
> | cross-entropy | 91.17 | 87.40 |
> | SEAL-dynamic-NCE | **91.53** | **88.11** |

---

### Meta-Review · Area_Chair_iDFC · 2022-08-31

**Recommendation:** Accept
**Confidence:** Certain

**Metareview:**

This work proposes using structured energy networks as loss functions for training feed forward networks to solve structured prediction tasks. The reviewers find the paper to be well written and easy to follow. The contribution is well positioned with respect to the literature and empirical results are strong. During the discussion period the authors addressed the concerns of the most negative reviewer sufficiently for them to increase their score. I can therefore recommend accepting this paper.

**Award:**

No

---

### Decision · Program_Chairs · 2022-09-14

Accept